# Attenuation of SARS-CoV-2 replication and associated inflammation by concomitant targeting of viral and host cap 2'-O-ribose methyltransferases

Valter Bergant[1], Shintaro Yamada[2], Vincent Grass[1], Yuta Tsukamoto[2], Teresa Lavacca[1], Karsten Krey[1], Maria-Teresa Mühlhofer[1], Sabine Wittmann[3], Armin Ensser[3], Alexandra Herrmann[3], Anja vom Hemdt[4], Yuriko Tomita[5], Shutoku Matsuyama[5], Takatsugu Hirokawa[6 7 8], Yiqi Huang[1], Antonio Piras[1], Constanze A Jakwerth[9], Madlen Oelsner[9], Susanne Thieme[10], Alexander Graf[10], Stefan Krebs[10], Helmut Blum[10], Beate M Kümmerer[4,11], Alexey Stukalov[1], Carsten B Schmidt-Weber[9], Manabu Igarashi[12,13], Thomas Gramberg[3], Andreas Pichlmair[1,14,*] & Hiroki Kato[2,*]

## Abstract

The SARS-CoV-2 infection cycle is a multistage process that relies on functional interactions between the host and the pathogen. Here, we repurposed antiviral drugs against both viral and host enzymes to pharmaceutically block methylation of the viral RNA 2'-O-ribose cap needed for viral immune escape. We find that the host cap 2'-O-ribose methyltransferase MTr1 can compensate for loss of viral NSP16 methyltransferase in facilitating virus replication. Concomitant inhibition of MTr1 and NSP16 efficiently suppresses SARS-CoV-2 replication. Using *in silico* target-based drug screening, we identify a bispecific MTr1/NSP16 inhibitor with anti-SARS-CoV-2 activity *in vitro* and *in vivo* but with unfavorable side effects. We further show antiviral activity of inhibitors that target independent stages of the host SAM cycle providing the methyltransferase co-substrate. In particular, the adenosylhomocysteinase (AHCY) inhibitor DZNep is antiviral in *in vitro*, in *ex vivo*, and in a mouse infection model and synergizes with existing COVID-19 treatments. Moreover, DZNep exhibits a strong immunomodulatory effect curbing infection-induced hyperinflammation and reduces lung fibrosis markers *ex vivo*. Thus, multispecific and metabolic MTase inhibitors constitute yet unexplored treatment options against COVID-19.

**Keywords** antivirals; COVID-19; host-directed; methyltransferase; SARS-CoV-2
**Subject Categories** Immunology; Microbiology, Virology & Host Pathogen Interaction
The EMBO Journal (2022) e111608

## Introduction

S-adenosylmethionine (SAM)-dependent methyltransferases (MTases) facilitate the transfer of a methyl group to a variety of substrates.

1 Institute of Virology, School of Medicine, Technical University of Munich (TUM), Munich, Germany
2 Institute of Cardiovascular Immunology, Medical Faculty, University Hospital Bonn (UKB), Bonn, Germany
3 Institute of Clinical and Molecular Virology, Friedrich-Alexander University Erlangen-Nürnberg, Erlangen, Germany
4 Institute of Virology, Medical Faculty, University of Bonn, Bonn, Germany
5 Department of Virology III, National Institute of Infectious Diseases (NIID), Tokyo, Japan
6 Transborder Medical Research Center, University of Tsukuba, Tsukuba, Japan
7 Division of Biomedical Science, University of Tsukuba, Tsukuba, Japan
8 Cellular and Molecular Biotechnology Research Institute, National Institute of Advanced Industrial Science and Technology, Tokyo, Japan
9 Center for Allergy & Environment (ZAUM), Technical University of Munich (TUM) and Helmholtz Center Munich, German Research Center for Environmental Health, Member of the German Center for Lung Research (DZL), CPC-M, Munich, Germany
10 Laboratory for functional genome analysis (LAFUGA), Gene Centre, Ludwig Maximilian University of Munich (LMU), Munich, Germany
11 German Center for Infection Research (DZIF), Bonn-Cologne Partner Site, Bonn, Germany
12 International Collaboration Unit, International Institute for Zoonosis Control, Hokkaido University, Sapporo, Japan
13 Division of Global Epidemiology, International Institute for Zoonosis Control, Hokkaido University, Sapporo, Japan
14 German Center for Infection Research (DZIF), Munich partner site, Munich, Germany
*Corresponding author. Tel: +49 89 4140 9270; E-mail: andreas.pichlmair@tum.de
**Corresponding author. Tel: +49 228287 51425; E-mail: hkato@uni-bonn.de
†These authors contributed equally to this work as first authors
‡These authors contributed equally to this work as second authors

Notably, mature mRNA from both humans and SARS-CoV-2 carry two distinct methylation marks at the 5′ end. The cap N7 methylation facilitates mRNA association with cap-binding proteins, which are essential for mRNA transport and translation (Muthukrishnan et al, 1978; Gebhardt et al, 2019). In addition, cap 2'O-ribose methylation is required by the virus to evade cell-intrinsic immunity, specifically from being sensed by the cellular pattern recognition receptors RIG-I (Schuberth-Wagner et al, 2015) and MDA5 (Züst et al, 2011) and restricted by the interferon (IFN)-induced protein IFIT1 (Daffis et al, 2010; Habjan et al, 2013; Abbas et al, 2017). SARS-CoV-2 encodes two viral MTases, non-structural protein (NSP) 14, a cap N7 MTase with proofreading exoribonuclease (ExoN) activity (Chen et al, 2009; Yan et al, 2021), and NSP16, a cap 2'O-ribose MTase (Decroly et al, 2008; Rosas-Lemus et al, 2020). NSP14 and NSP16 were so far believed to be the sole MTases involved in their respective steps of viral RNA maturation. Therefore, both enzymes were considered pivotal for virus replication and recognized as potential targets for anti-SARS-CoV antiviral therapies (Decroly et al, 2008; Chen et al, 2009). In particular, the activity of NSP16 was shown to be required for IFN resistance and virulence of related SARS and MERS coronaviruses in an MDA5- and IFIT1-dependent manner (Menachery et al, 2014, 2017). Despite recent structural insights, specific targeting of viral MTases remains challenging (Chen et al, 2011; Rosas-Lemus et al, 2020; Ahmed-Belkacem et al, 2020; Vijayan et al, 2020; Aldahham et al, 2020).

In humans, more than 150 SAM-dependent MTases contribute to a plethora of biological processes. Of particular importance is their involvement in epigenetic gene regulation via histone H3K27 methylation, a repressive chromatin mark deposited by polycomb repressive complex 2 (PRC2), which has been linked to disease-relevant processes such as tissue fibrosis (Xiao et al, 2016) and innate immune responses (Chen et al, 2013; Arbuckle et al, 2017). Inhibition of the enhancer of zeste 2 PRC2 subunit (EZH2) was shown to reduce TGF-β1-induced human lung fibroblast-to-myofibroblast transformation and to attenuate bleomycin-induced pulmonary fibrosis in mice (Xiao et al, 2016). Moreover, it was also associated with reduction in NF-kB-dependent responses via upregulation of NF-kB inhibitors TNFAIP3/A20 and NFKBIA (Loong, 2013) and activation of the IFN response (Wee et al, 2014; Morel et al, 2021). Notably, NF-kB signaling is highly active in SARS-CoV-2-infected cells and in COVID-19 patients, thereby contributing to virus-induced immunopathology (Leisman et al, 2020). At the same time, SARS-CoV-2 is strongly inhibited by the antiviral functions invoked by type I IFN signaling but a number of viral proteins actively perturb this pathway at multiple levels (Miorin et al, 2020; Stukalov et al, 2021). Direct or indirect inhibition of MTase EZH2 could therefore lead to a reduction in lung fibrosis and relieve cytokine imbalance, both associated with negative disease outcomes, and thereby contribute to the resolution of acute and long-term effects of COVID-19.

One-carbon metabolism, and in particular the S-adenosylmethionine (SAM) cycle, is essential for maintaining the activity of SAM-dependent MTases. The SAM cycle produces the universal methyl group donor SAM and recycles the S-adenosylhomocysteine (SAH), which is a product inhibitor of SAM-dependent MTases (Hoffman et al, 1980). The SAM cycle can be subdivided into four enzymatic steps: the methionine biosynthesis, SAM biosynthesis, SAM-dependent methylation of substrates, and SAH hydrolysis. Of these steps, three rely on host metabolic enzymes and can be perturbed by host-targeting inhibitors, while SAM-dependent methylation is driven by distinct MTases, which are challenging to target specifically. The host metabolic enzymes involved in the SAM cycle are the methionine synthases (BHMT, BHMT2, and MTR together with the factor required for its regeneration MTRR), methionine adenosyltransferases (MAT1A, MAT2A, and associated regulator MAT2B), and adenosylhomocysteinase AHCY. Pharmaceutical targeting of the SAM cycle at different stages is a potential treatment option for a number of cancers (Uchiyama et al, 2017; Hasan et al, 2019; Konteatis et al, 2021) and was shown to be well tolerated in model organisms (Sun et al, 2015; Konteatis et al, 2021). Inhibitors of the SAM cycle enzymes negatively influence key cellular methylation capacity biomarkers, i.e., reduce the levels of SAM, increase the levels of SAH, and reduce the SAM-to-SAH ratio (Collinsova et al, 2006; Strakova et al, 2011; Aury-Landas et al, 2019), leading to broad-spectrum inhibition of SAM-dependent MTases through substrate starvation and product inhibition.

Drug repurposing is the most rapid antiviral drug development approach (Kaufmann et al, 2018; Garća-Serradilla et al, 2019; Chitalia & Munawar, 2020). Host-directed antiviral drug repurposing is, in particular, attractive because it leverages a larger set of well-defined drugs used for treating non-infectious diseases and limits the risk to select for viral escape mutants. It allows for synergistic use of the state-of-the-art knowledge of both virus and host biology and has the potential for developing cross-functional and broad-spectrum antivirals. Targeting known disease-promoting factors, i.e., target-based host-directed drug repurposing, led to the discovery of the host protease inhibitor camostat (Kawase et al, 2012) and inosine-5′-monophosphate dehydrogenase (IMPDH) inhibitors ribavirin and VX-497 (Markland et al, 2000). Such approaches, often based on in silico screens, are of specific importance in tackling emerging and pandemic viruses and viral families for which extensive molecular characterization, otherwise serving as the basis for developing direct-acting antivirals, is incomplete or missing.

Herein, we leveraged both direct-acting and host-directed antiviral drug repurposing to explore the antiviral potential of pharmaceutical inhibition of SARS-CoV-2 cap 2'-O-ribose methyltransferase NSP16. Through in silico molecular docking, we identified a set of drug candidates with the potential to inhibit MTase activity of NSP16. While the inhibitor tubercidin (7-deazaadenosine) proved to be highly antiviral against SARS-CoV-2, other inhibitors with comparable or higher docking scores did not significantly affect the virus replication. In line with previous observations for SARS-CoV (Menachery et al, 2014), we show that genetically inflicted loss of function of NSP16 results in only moderate attenuation of SARS-CoV-2, indicating that stand-alone inhibition of NSP16 is insufficient to impair virus replication. Surprisingly, SARS-CoV-2 NSP16 mutant virus failed to replicate in cells that were depleted for the host cap 2'O-ribose MTase MTr1 (CMTR1, FTSJD2; Bélanger et al, 2010), suggesting that this host protein can compensate for the activity of its viral analog NSP16. Indicative of promiscuity, tubercidin potently inhibited both NSP16 and MTr1 in vitro, further emphasizing that a concomitant inhibition of NSP16 and MTr1 is pivotal for effective antiviral treatment. The activity of MTr1 and NSP16 critically depends on the metabolite homeostasis maintained by the host SAM cycle. We further explored the antiviral potential of host-directed SAM cycle inhibitors (SCIs), which in an indirect manner induce a

metabolic broad-spectrum MTase inhibition. We show that targeting all three independent enzymatic steps of the SAM cycle by unrelated small molecule inhibitors significantly reduces SARS-CoV-2 proliferation *in vitro*. Notably, the SAM cycle inhibitor 3-deazaneplanocin A (DZNep), an inhibitor of AHCY, has especially potent and selective antiviral efficacy against SARS-CoV-2 in *in vitro*, in *ex vivo*, and in a mouse infection model. In line with its known facilitative effect on tissue repair, DZNep treatment of primary human lung cells exhibited a strong immunomodulatory effect curbing infection-induced hyperinflammation and reduced lung fibrosis- and coagulopathy-related markers. Moreover, our data demonstrate that DZNep synergizes with the current treatment options remdesivir and interferon-alpha. These findings show that targeting the MTases involved in SARS-CoV-2 viral life cycle is a novel and therapeutically viable antiviral strategy for treatment of COVID-19.

## Results

### *In silico* screening identified NSP16 inhibitors with potent anti-SARS-CoV-2 activity

We employed a target-based drug repurposing approach aimed toward identification of potential novel NSP16 inhibitors. In particular, we utilized *in silico* screening of 4,991 unique DrugBank compounds for binding to the SAM-binding pocket of the SARS-CoV-2 NSP10/16 complex (PDB 6W4H; Fig 1A). As expected, SAM and SAH had the highest docking scores in our screen, followed by the SAM analog sinefungin (Krafcikova *et al*, 2020) and numerous other adenosine mimics (Dataset EV1). Based on the results of the *in silico* screen, we shortlisted 14 commercially available compounds (Fig 1B) and tested them for antiviral efficacy against SARS-CoV-2. Toward this, we pretreated human lung-derived cell line A549 complemented with the SARS-CoV-2 receptor angiotensin-converting enzyme 2 (A549-ACE2) with selected compounds at 1 µM concentration and infected them with SARS-CoV-2 at MOI 0.01. Twenty-four hours post-infection, RNA was isolated and the abundance of viral transcript encoding envelope protein (*E*) quantified by RT–qPCR. Surprisingly, most compounds did not exhibit antiviral activity with a notable exception of tubercidin, which was found to be highly potent under conditions used (Fig 1C). The *in silico* docking screen suggested that tubercidin binds to the SAM-binding pocket of NSP16 (Fig 1D), indicating that it may serve as a potential inhibitor of its cap 2'O-ribose MTase activity. We employed an *in vitro* MTase activity assay to experimentally test whether tubercidin influences the enzymatic activity of the NSP10/16 complex. Toward this, we used *in vitro*-transcribed cap0 RNA as the methyl group recipient and measured the MTase activity of recombinant NSP16/10 by quantifying the transferred tritium-labeled methyl groups from SAM[$^3$H]. While only mildly inhibiting unrelated Vaccinia virus MTase VP39, tubercidin significantly reduced the enzymatic activity of the MTase NSP10/16 (Fig 1E), indicating specificity in this assay.

To further explore the antiviral efficacy of tubercidin (Schultz *et al*, 2022), we pretreated A549-ACE2 cells with tubercidin at a range of concentrations 3 h prior to infection with SARS-CoV-2 and quantified SARS-CoV-2 nucleoprotein (N) accumulation by Western blot and immunostaining. In agreement with our previous findings, we observed a strong reduction of SARS-CoV-2 N accumulation in tubercidin-treated conditions as compared to control treatments (Figs 1F and EV1A). In an analogous experiment with SARS-CoV, we observe a similar trend, indicating that tubercidin is antiviral against both highly related coronaviruses (Fig EV1B and C). We further employed liquid chromatography coupled to tandem mass spectrometry (LC–MS/MS) analysis to evaluate abundance changes of viral proteins upon tubercidin or vehicle pretreatment of SARS-CoV or SARS-CoV-2-infected A549-ACE2 cells. We observed a prominent and highly significant tubercidin-dependent reduction in accumulation across all viral proteins (Fig 1G and Dataset EV2). We additionally observe a potent reduction in levels of viral RNA in SARS-CoV-2- or SARS-CoV-infected and tubercidin-treated A549-ACE2 cells as compared to vehicle-treated controls (Fig EV1D and E). Moreover, the production of infectious viral progeny (Figs 1H and EV1F) and viral RNA accumulation (Fig EV1G) was strongly reduced in the supernatants of tubercidin-treated SARS-CoV-2-, SARS-CoV-, or MERS-CoV-infected cells as compared to control treatments.

Next, we examined potential variability between antiviral efficacies of tubercidin against the different SARS-CoV-2 variants of concern. Toward this, we pretreated A549-ACE2 cells with 1 µM tubercidin and infected them with variants of concern alpha (B.1.1.7), beta (B.1.351), and delta (B.1.617.2) at MOI 0.01 for 24 h. For all viruses tested, we observed a comparable tubercidin-dependent reduction in viral RNA accumulation (Fig 1I). Taken together, employing target-based drug repurposing we identified tubercidin as a novel inhibitor of SARS-CoV-2 NSP16 with a potent antiviral efficacy against SARS-CoV-2 and other tested betacoronaviruses.

### Concomitant inhibition of NSP16 and MTr1 is necessary for efficient suppression of SARS-CoV-2

NSP16 was previously proposed to be critical for SARS-CoV replication (Decroly *et al*, 2008) and was found to be required for IFN resistance and virulence of related SARS-CoV (Menachery *et al*, 2014) and MERS-CoV (Menachery *et al*, 2017). To assess the functional role of NSP16 in SARS-CoV-2 replication, we generated a mutant SARS-CoV-2 with the functionally deficient NSP16 harboring D130A K170A mutations (designated SARS-CoV-2 NSP16mut). These mutations abrogate NSP16 MTase activity (manuscript by T. Gramberg in preparation). To explore the effect of NSP16 deficiency, we monitored virus propagation levels in Calu-3 cell supernatants over a 6-day period. We observed only a minor loss in replication competency of SARS-CoV-2 NSP16mut compared with the wild-type (wt) SARS-CoV-2 (Fig 2A), indicating a prominent but not vital role of NSP16 in SARS-CoV-2 replication. An analogous observation was previously reported for SARS-CoV (Menachery *et al*, 2014). We hypothesized that the potent activity of tubercidin (Fig 1C and F–I) may be due to additional targeting of host factors that compensate for the loss of NSP16 activity. A potential host target of tubercidin is the cellular cap 2'O-ribose MTase MTr1, which is active on the host RNA. Notably, confocal imaging indicated nuclear and cytoplasmic localization of MTr1 in A549 cells, which is also in line with reports in public repositories (Williams *et al*, 2020). MTr1 expression is upregulated by IFN-α treatment (Williams *et al*, 2020; Fig EV1H). Nucleocytoplasmic fractionation of A549 cells further confirmed

cytoplasmic localization of MTr1 and increase in cytoplasmic MTr1 abundance upon IFN-β treatment (Fig EV1I). To evaluate whether tubercidin targets MTr1, we conducted molecular docking simulations, which indeed indicated that tubercidin can bind to the active site of MTr1 (Fig 2B). Moreover, tubercidin inhibited MTr1 function *in vitro* (manuscript in preparation by HK), potentially leading to concomitant inhibition of MTr1 and NSP16 in tubercidin-treated SARS-CoV-2-infected cells.

To assess the potential role of MTr1 in the SARS-CoV-2 life cycle, we tested to what extent MTr1-deficient A549-ACE2 cells (MTr1 KO) can support SARS-CoV-2 replication in comparison with controls. Toward this, we first characterized MTr1 KO cells in uninfected and

infected conditions. Replication of a model virus (vesicular stomatitis virus, VSV) was not affected by the MTr1 KO (Fig 2C). In line with the literature (Williams *et al*, 2020), the basal and VSV-induced IFN-β mRNA levels were similar in mock- and virus-infected control and MTr1 knockout cells, respectively (Fig 2C). However, we unexpectedly observed a major impairment in SARS-CoV-2 protein and RNA accumulation in MTr1 KO cells as compared to non-targeting control cells (Figs 2D and EV1J). In addition, we could observe virus-induced cytopathic effects in control cells but not in MTr1 KO cells (Fig EV1K). Most notably, release of infectious SARS-CoV-2 NSP16mut was almost undetectable in MTr1 KO cells (Fig 2E). These data indicated that human MTr1 serves as a host

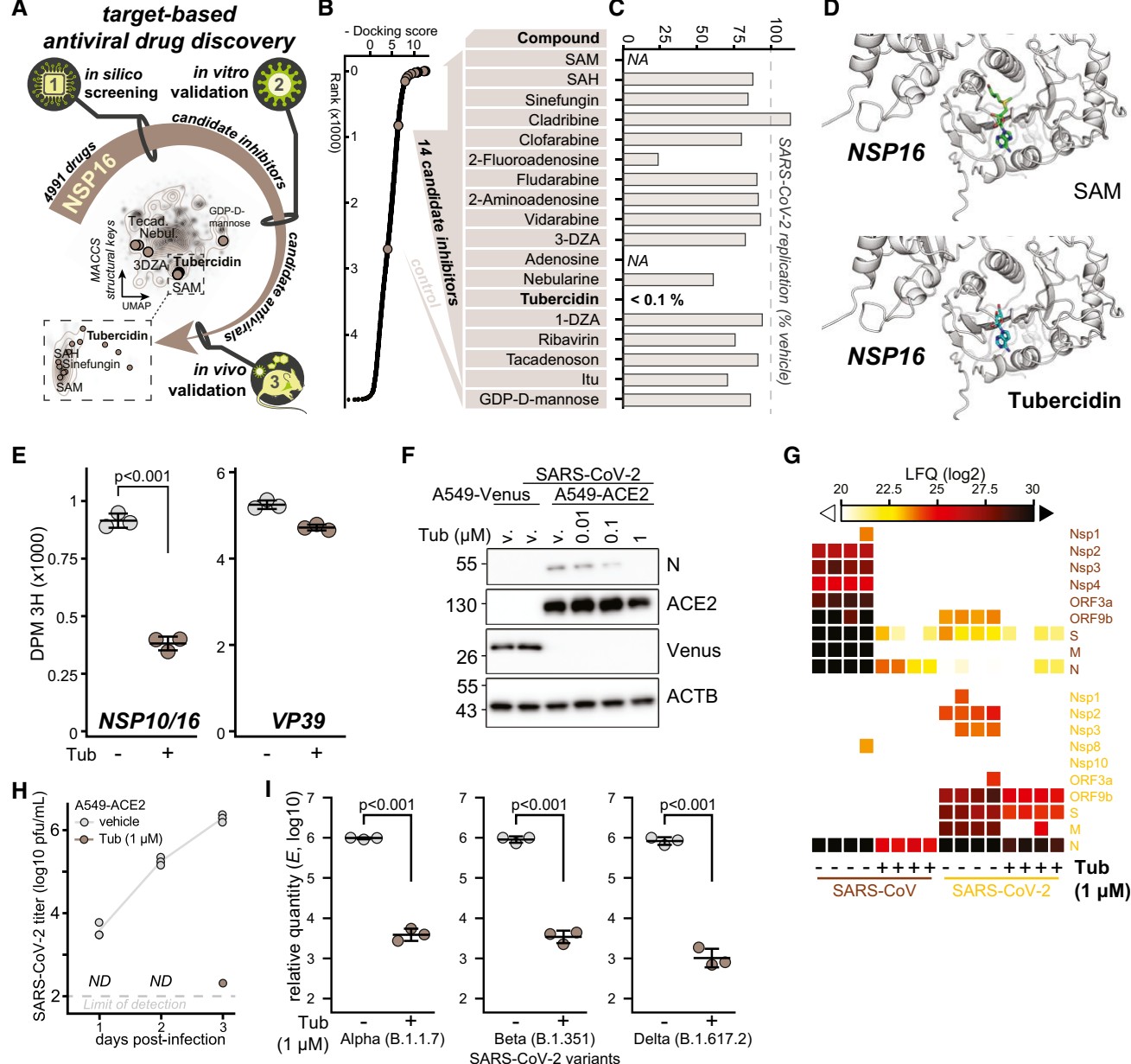

**Figure 1.**

◀

**Figure 1.** *In silico* screening identified NSP16 inhibitors with potent anti-SARS-CoV-2 activity.

A   Schematic representation of the target-based antiviral drug discovery pipeline employed herein. A total of 4,991 chemical compounds from DrugBank were docked to the SAM-binding pocket in the crystal structure of SARS-CoV-2 NSP16 (PDB ID: 6W4H), obtaining 14 commercially available compounds with high docking score that were used, along a control, in an *in vitro* antiviral assay. UMAP dimensionality reduction according to MACCS structural keys. Plots depict structural diversity of the shortlisted compounds alongside the compounds used in the *in silico* screen (contour lines), overlaid on the top of density distribution of ~800,000 bioactive small molecules (Duran-Frigola *et al*, 2020).

B   Docking score from the *in silico* screen, depicted for all (black) and shortlisted (beige, table) compounds (full list provided in Dataset EV1).

C   Results of *in vitro* antiviral assay for compounds according to (B). A549-ACE2 cells were pretreated with indicated compounds at 1 μM concentration for 3 h prior to infection with SARS-CoV-2 at MOI 0.01. Twenty-four hours post-infection, expression of SARS-CoV-2 transcript encoding envelope protein (*E*) was quantified by RT–qPCR as a measure of SARS-CoV-2 replication and is shown as a percent of vehicle-treated control. *NA*, not assayed.

D   The docking poses of SAM (top) or tubercidin (bottom) in the SAM-binding pocket of SARS-CoV-2 NSP10/16 (PDB 6W4H).

E   Disintegrations per minute (DPM 3H) originating from *in vitro*-transcribed cap0 RNA methylated by the NSP10/16 complex (left) or Vaccinia virus VP39 (right) with optional addition of 10 mM tubercidin. SAM[$^3$H] was provided as a substrate. Error bars correspond to mean ± SD of three reaction replicates; statistics were calculated using Student's two-sided *t*-test between indicated conditions.

F   A549-ACE2 or control A549-Venus cells were pretreated with tubercidin or vehicle (DMSO) 3 h prior to infection with SARS-CoV-2 at MOI 0.1. After 24 h, the abundance of SARS-CoV-2 nucleoprotein (N), ACE2, Venus, and β-actin (ACTB, loading control) was visualized using Western blotting. The presented data are representative of three independent repeats.

G   A549-ACE2 cells were pretreated with tubercidin or vehicle (DMSO) 3 h prior to infection with SARS-CoV (MOI 0.01) or SARS-CoV-2 (MOI 0.1). After 24 h, protein content of the cells was isolated and subjected to LC–MS/MS-based proteomics analysis. Label-free quantification (LFQ)-based abundance of detected viral proteins is depicted.

H   A549-ACE2 cells were pretreated with tubercidin or vehicle (DMSO) 3 h prior to infection with SARS-CoV-2 at MOI 0.01. At 1 h post-infection, medium change was performed. At the indicated days post-infection, infectious viral progeny was quantified in the supernatants from three independently infected wells by plaque assay on Vero cells. *ND*, not detected. The measurements are representative of two independent repeats.

factor in the absence of a functional viral 2'O methyltransferase. To test whether this was specific to SARS-CoV-2 or whether this phenotype can also be observed for other viruses that lack their 2'O methyltransferase activity, we infected wt and MTr1 knockout cells with wt yellow fever virus (YFV) and YFV with a mutation in its 2'O-ribose MTase (YFV-E218A; Zhou *et al*, 2007). Notably, both wt and mutant YFV grew to similar titers in wt and MTr1-deficient cells, indicating that YFV does not rely on cellular MTr1 (Fig 2F). We concluded that MTr1 is a SARS-CoV-2-specific host factor with a redundant or cooperative function to NSP16. These genetic observations further indicated that a concomitant pharmaceutical inhibition of viral NSP16 and host MTr1 is critical for the efficient suppression of SARS-CoV-2.

In order to assess whether tubercidin is antiviral against SARS-CoV-2 *in vivo*, we infected C57BL/6 mice with SARS-CoV-2 beta variant (B.1.351, 250 pfu intranasal) and treated the animals at the day (day 0) and 1 day after infection (day 1) with tubercidin (25 μg, intranasal application; Fig 2G). At day 2 post-infection, which represented the early acute stage of infection, the animals were sacrificed and lungs were harvested to quantify the viral load. We observed a significant reduction of the viral RNA level in the lungs of tubercidin-treated animals relative to the control (Fig 2H). However, we also observed significant weight loss of treated mice (Fig 2I), suggesting *in vivo* toxicity of this compound that may be prohibitive for its clinical application. Taken together, we identified MTr1 as a novel SARS-CoV-2 host factor with a redundant or cooperative activity to the viral MTase NSP16. We further showed that concomitant targeting of both NSP16 and MTr1 is critical for efficient repression of SARS-CoV-2 replication. While dual targeting of NSP16 and MTr1 may be a central property explaining the antiviral efficacy of tubercidin observed *in vitro* and *in vivo*, alternative targeting strategies are required to circumvent its toxicity.

## SAM cycle enzymes are key host factors facilitating SARS-CoV-2 proliferation

We explored alternative strategies of concomitant inhibition of NSP16 and MTr1 that may be applicable for clinical settings. The activity of both NSP16 and MTr1 is influenced by the levels of their substrate S-adenosylmethionine (SAM) and product inhibitor S-adenosylhomocysteine (SAH). Homeostasis of both SAM and SAH is solely driven by the enzymes of the host SAM cycle (Fig 3A). Inhibition of the SAM cycle enzymes causes a metabolic broad-spectrum MTase inhibition through substrate starvation and product inhibition (Hoffman *et al*, 1980). This may exert an antiviral effect against SARS-CoV-2 mechanistically similar to tubercidin (Fig 3B). To explore the role of the SAM cycle enzymes in SARS-CoV-2 infection, we used CRISPR/Cas9 to genetically ablate MAT2A, the main methionine adenosyltransferase of extrahepatic tissues, and AHCY, the sole human adenosylhomocysteinase, in A549-ACE2 cells. We employed time-resolved live-cell fluorescent imaging to evaluate cell growth and proliferation of GFP-expressing SARS-CoV-2 reporter virus (SARS-CoV-2-GFP; Thi Nhu Thao *et al*, 2020; Stukalov *et al*, 2021). Cells lacking MAT2A or AHCY exhibited minor reduction in cell growth relative to non-targeting control cells (NTC; Fig EV2A and B). Notably, compared with NTC, targeting MAT2A and AHCY significantly restricted SARS-CoV-2 replication (Fig 3C).

We evaluated anti-SARS-CoV-2 efficacy of the inhibitor of methionine synthases BHMT/BHMT2 (CBHcy; Jiracek *et al*, 2006), inhibitors of methionine adenosyltransferases MAT1A/MAT2A/MAT2B (MI1 (Konteatis *et al*, 2018), FIDAS-5 (Zhang *et al*, 2013; Sviripa *et al*, 2014), and PF-9366 (Quinlan *et al*, 2017)), and inhibitors of adenosylhomocysteinase AHCY (DZNep (Glazer *et al*, 1986) and DER (Schanche *et al*, 1984)), collectively termed SAM cycle inhibitors (SCIs; Fig 3A). Notably, the inhibition of all SAM cycle

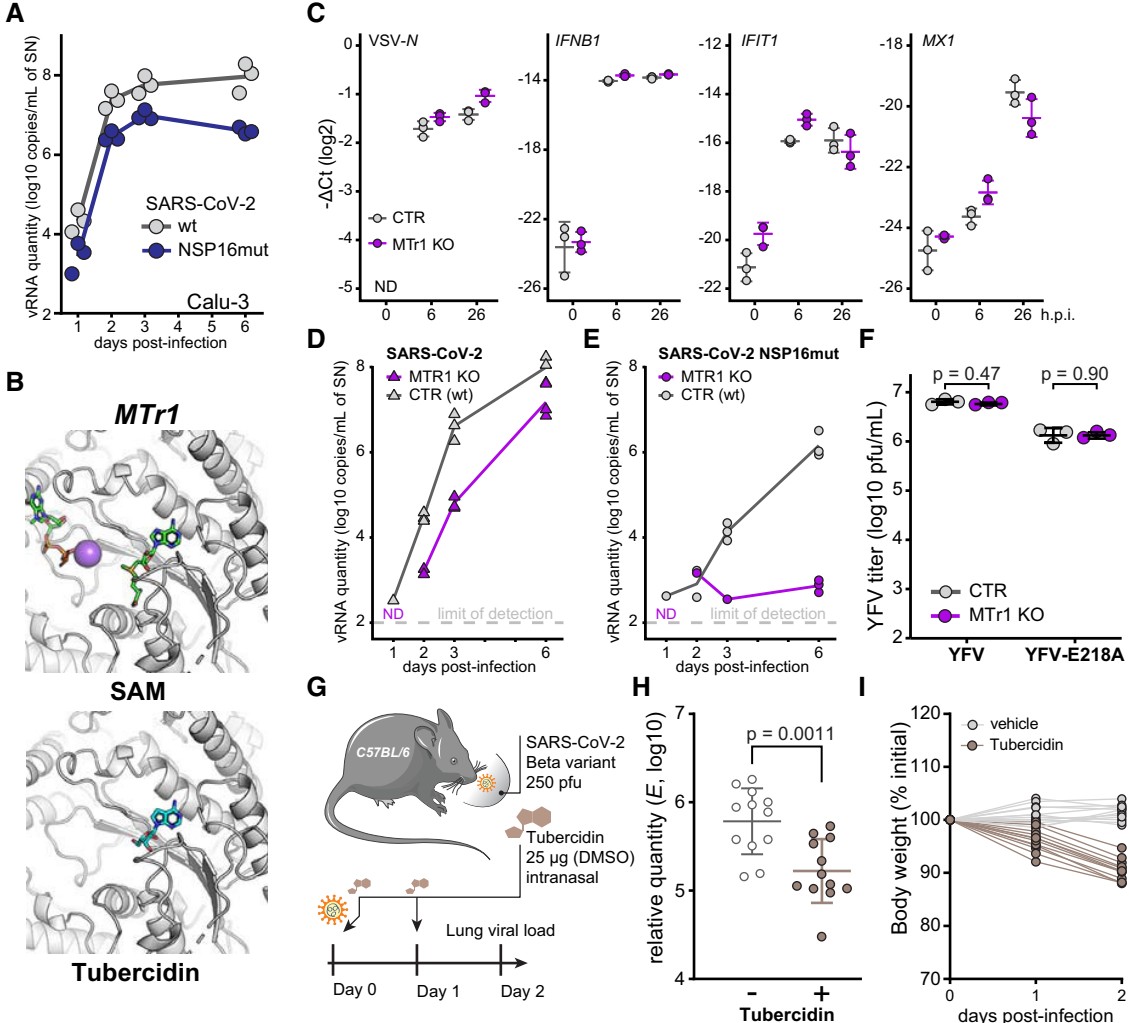

**Figure 2. Concomitant inhibition of NSP16 and MTr1 is necessary for efficient suppression of SARS-CoV-2.**

A    Calu-3 cells were infected with SARS-CoV-2 or SARS-CoV-2 NSP16mut at 5,000 RNA copies/well. At indicated times post-infection, RNA from the supernatants from *n* = 3 independently infected wells was isolated and RT–qPCR used to quantify the presence of viral RdRp encoding RNA.

B    The docking poses of SAM (top) or tubercidin (bottom) in the SAM-binding pocket of human MTr1 (PDB 4N49).

C    CTR or MTr1 KO A549-ACE2 cells were infected with VSV-GFP at MOI 1 or mock (0 h post-infection, h.p.i.). At indicated times post-infection, depicted transcript abundance was quantified by RT–qPCR (relative to 18S rRNA). Error bars correspond to mean ± SD of *n* = 3 independently infected wells.

D, E  Control (CTR) or MTr1 knockout (KO) A549-ACE2 cells were infected with SARS-CoV-2 (D) or SARS-CoV-2 NSP16mut (E) at 5,000 RNA copies/well. At indicated times post-infection, RNA from the supernatants from *n* = 3 independently infected wells was isolated and RT–qPCR used to quantify the presence of viral RNA targeting RdRp coding region. Dotted line—not detected.

F    CTR or MTr1 KO A549-ACE2 cells were infected with YFV or YFV-E218A at MOI 0.1. At 2 days post-infection, infectious viral progeny was quantified in the supernatants by plaque assay on Vero cells. Error bars correspond to mean ± SD of *n* = 3 independently infected wells.

G    Schematic representation of the *in vivo* antiviral assay employing a murine infection model.

H, I  C57BL/6 mice were infected with SARS-CoV-2 beta variant (250 pfu, intranasal) and treated at D0 and D1 with tubercidin (25 µg, intranasal). Forty-eight hours post-infection, lungs of infected mice were isolated. The presented data were pooled from two independent experiments. (H) Abundance of SARS-CoV-2 transcript *E* was quantified in the lung samples by RT–qPCR as a measure of lung viral load. Mean ± SD of *n* = 12 animals per condition is shown; statistics were calculated using Student's two-sided *t*-test as indicated. (I) Animal body weight, depicted as percentage of initial weight, measured at indicated times post-infection.

enzymes exhibited a significant antiviral effect against SARS-CoV-2 (Figs 3D and EV2C–G). While most inhibitors showed significant antiviral effects at µM concentrations, DZNep, an AHCY inhibitor, was most potent and led to a significant reduction in SARS-CoV-2 growth in the nM range (Fig 3D and E). In contrast to tubercidin (Figs 3F and G, and EV2H), treatment with SCIs had minor-to-no impact on cell proliferation for most compounds (Figs 3E and

EV2C–G), indicating that the observed antiviral effect for those compounds was not due to altered cellular viability or growth rates. To corroborate these findings, we evaluated the antiviral efficacy of DZNep, FIDAS-5 and CBHcy in Vero E6 cells. Toward this, we pretreated Vero E6 cells with SCIs at different concentrations, infected them with wild-type (wt) SARS-CoV-2, and after 48 h quantified the amount of released viral progeny in the supernatant by plaque

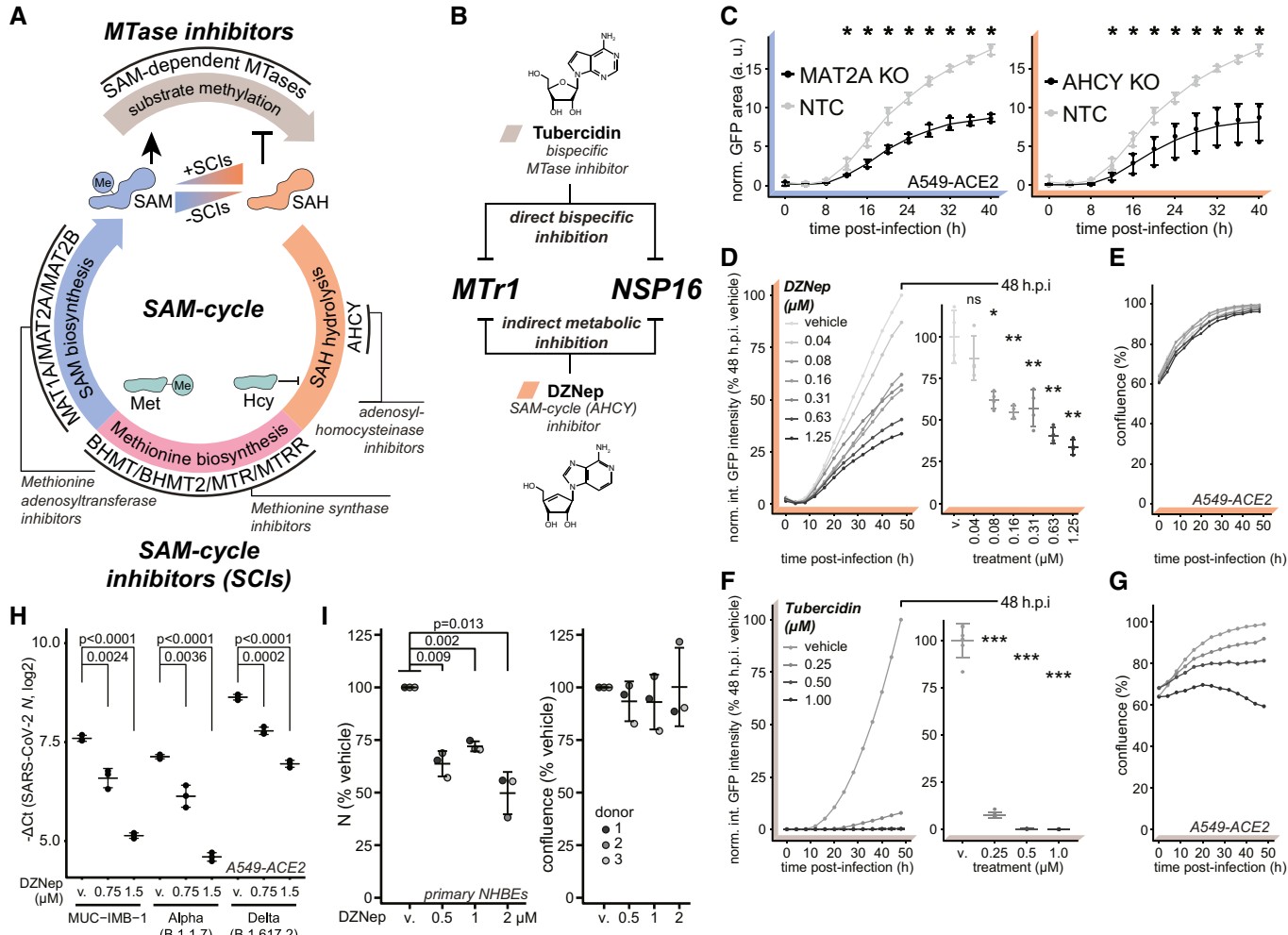

**Figure 3. SAM cycle enzymes are key host factors facilitating SARS-CoV-2 proliferation.**

A   Schematic representation of the SAM cycle, metabolites, enzymatic components, and inhibitors thereof.

B   Schematic representation of the two orthogonal mechanisms allowing for concomitant inhibition of MTases NSP16 and MTr1.

C   AHCY KO, MAT2A KO, or non-targeting control (NTC) A549-ACE2 cells were infected with SARS-CoV-2-GFP at MOI 3 and normalized GFP area plotted over time as a measure of reporter virus growth. Error bars correspond to mean ± SD of *n* = 3 independently infected wells. Statistics were calculated using Student's two-sided *t*-test between individual KOs and NTC at indicated times post-infection. ns *P* > 0.05, * *P* < 0.05.

D–G   The effect of the SAM cycle inhibitor DZNep and bispecific MTase inhibitor tubercidin on cell and virus growth. A549-ACE2 cells were pretreated for 6 h with indicated concentrations of (D, E) DZNep or (F, G) tubercidin and infected with SARS-CoV-2-GFP at MOI 3. Normalized integrated GFP intensity and confluence are depicted as a measure of virus replication and cell growth, respectively. Error bars correspond to mean ± SD of (D, E) *n* = 4 and (F, G) *n* = 6 independently infected wells; the measurements are representative of three independent repeats. Statistics were calculated using Student's two-sided *t*-test between indicated treatment concentrations and respective vehicle controls (v., DZNep—PBS; tubercidin—DMSO). ns *P* > 0.05, * *P* < 0.05, ** *P* < 0.01, *** *P* < 0.001.

H   A549-ACE2 cells were pretreated for 6 h with indicated concentrations of DZNep or vehicle (v., PBS) and infected with indicated variants of SARS-CoV-2 at MOI 3 for 24 h. Graph shows *N* mRNA expression normalized to housekeeping gene (*RPLP0*); error bars represent mean ± SD of *n* = 3 independently infected wells. *P*-values were calculated using Student's two-sided *t*-test as indicated.

I   NHBEs derived from three independent donors were pretreated for 6 h with indicated concentrations of DZNep or vehicle and infected with SARS-CoV-2 for 24 h. Cells were fixed, and the abundance of SARS-CoV-2 N was quantified by immunofluorescent staining. Shown are vehicle-normalized integrated anti-N fluorescent intensity and cell confluence; error bars represent mean ± SD of *n* = 3 donors. Statistics were calculated using one sample Student's two-sided *t*-test.

Source data are available online for this figure.

assay. In line with the reporter virus assays, we observed dose-dependent suppression of the wt SARS-CoV-2 for the tested SCIs (Fig EV3A–C). Interestingly, while DZNep, an inhibitor of AHCY, proved the most efficacious in the reporter virus assay, CBHcy, an inhibitor of BHMT/BHMT2, exhibited the highest antiviral effect in this setting (Fig EV3C).

DZNep was previously shown to have antiviral activity against some viruses but not against others (Tseng *et al*, 1989; Chen *et al*, 2013; Arbuckle *et al*, 2017). We first explored the antiviral effect of DZNep against the early clinical isolate SARS-CoV-2-MUC-IMB-1, the alpha (B.1.1.7) and the delta (B.1.617.2) SARS-CoV-2 variants. The reduction of SARS-CoV-2 *N* mRNA levels as a proxy for antiviral

efficacy of DZNep against the tested variants was comparable (Fig 3 H), indicating that they are similarly susceptible to AHCY inhibition. DZNep was shown to be ineffective in reducing SARS-CoV lung titer in a murine infection model (Barnard *et al*, 2006). We employed a Western blot-based readout to compare antiviral efficacy against SARS-CoV-2 and SARS-CoV. In agreement with our previous findings, we observed a reduction in SARS-CoV-2 N accumulation in DZNep-treated conditions (Fig EV3D). However, in contrast to tubercidin but in line with the literature (Barnard *et al*, 2006), we observed no clear effect of DZNep on SARS-CoV N accumulation under the tested conditions (Fig EV3E).

In order to explore whether DZNep treatment impairs virus replication or an earlier process such as virus entry, we compared SARS-CoV-2-GFP reporter virus growth curves upon treatment of A549-ACE2 cells with DZNep, IFN-α, or neutralizing antisera (Lainšček *et al*, 2021). While antisera, which reduces virus infection rates, delayed onset of virus replication by 3–4 h, it did not affect the overall increase in GFP signal over time (Fig EV3F, left). In contrast, IFN-α treatment restricts virus replication at multiple levels downstream of viral entry, which is characterized by reduced maximal virus proliferation rate and a tilted slope in GFP signal (Fig EV3F, middle). Notably, SARS-CoV-2 reporter virus growth rates in DZNep-treated cells did not delay onset of virus replication but were comparable to growth rates obtained in IFN-α-treated cells (Figs 3D and EV3F, right). To further corroborate these findings, we treated A549-ACE2 cells with DZNep 4 h prior, at the time of, and 4 h post-infection. We detected no major differences in its antiviral efficacy (Fig EV3G), indicating that inhibition of viral entry is not the main driver of antiviral efficacy of DZNep but that a post-entry process is affected by DZNep.

DZNep was shown to be highly efficacious against Ebolavirus infection *in vivo* (Bray *et al*, 2000, 2002), in the context of which it strongly stimulated type I IFNs (Bray *et al*, 2002). To explore the contribution of the IFN response to antiviral efficacy of DZNep against SARS-CoV-2, we used a STAT1-deficient A549-ACE2 cell line and compared its response with that of DZNep relative to the NTC with optional IFN-α co-treatment (Fig EV3H). As expected, treatment of NTC cells with IFN-α significantly attenuates virus growth (Mantlo *et al*, 2020), as did treatment of NTC cells with DZNep. Interestingly, co-treatment with DZNep and IFN-α led to further reduction in virus propagation, suggesting that IFN-α may potentiate antiviral efficacy of DZNep. In line with our observations from IFN-deficient Vero E6 cells, IFN-α was no longer active in STAT1-deficient cells, while DZNep retained its antiviral activity (Fig EV3H).

DZNep was previously shown to invoke depletion of H3K27 trimethylation in cancer cells (Tan *et al*, 2007; Miranda *et al*, 2009), suggesting inhibitory activity on the MTase EZH2, the enzymatic component of the PRC2 complex. It is possible that SCIs, due to their related mode of action, in general confer EZH2 inhibition and subsequently deplete H3K27 trimethylation levels. We used tazemetostat (Knutson *et al*, 2014), a potent and selective competitive inhibitor of EZH2 approved for treatment of epithelioid sarcoma, to explore the antiviral potential of stand-alone EZH2 inhibition against SARS-CoV-2. We observed a moderate tazemetostat-dependent decrease in virus proliferation (Fig EV3I), suggesting that EZH2 inhibition may partially contribute to the antiviral efficacy of DZNep against SARS-CoV-2.

In order to corroborate our *in vitro* findings, we employed primary normal human bronchial epithelial cells (NHBEs) as a highly relevant lung-derived *ex vivo* infection model. Toward this, we pretreated NHBEs with various concentrations of DZNep, infected them with SARS-CoV-2 at MOI 3 for 24 h and quantified viral N accumulation by immunofluorescence analysis. In line with observations in cell lines, DZNep treatment mediated a significant decrease in abundance of N in human primary cells (Fig 3I). Collectively, this shows that the SAM cycle enzymes are key host factors for SARS-CoV-2 replication that can be pharmacologically targeted to exert an antiviral effect.

## DZNep treatment modulates tissue and immune processes

In order to explore the effect of DZNep as antiviral SCI on host and viral protein expression, we employed LC–MS/MS analysis. In particular, we evaluated protein abundance changes upon DZNep or vehicle pretreatment of mock-, SARS-CoV- or SARS-CoV-2-infected A549-ACE2 cells and NHBEs (Fig 4A, and Datasets EV3 and EV4). We quantified abundance of 5,957 and 6,129 proteins in A549-ACE2s and NHBEs, respectively, and evaluated the effect of SARS-CoV and SARS-CoV-2 infection, as well as drug treatment using the LASSO statistical model (Figs 4A and EV4A). In both A549-ACE2 and NHBEs, SARS-CoV-2 and SARS-CoV infections, as well as treatment in distinct conditions, elicited comparable proteome changes (Figs 4B and EV4B–E). Consistent with our previous findings, we show DZNep-dependent inhibition of SARS-CoV-2 but not of SARS-CoV, as determined by abundance changes of N and spike (S) proteins (Fig 4C). In NHBEs, we observed infection-dependent upregulation of proteins associated with innate immunity, which was further amplified by DZNep treatment and which may contribute to the antiviral activity of DZNep (Fig EV4E). Interestingly, we also observed DZNep-dependent upregulation of numerous SAM-dependent MTases, in A549-ACE2 cells (e.g., NSUN2, NOP2, METTL3, CMTR2, NTMT1, and FTSJ1; Fig EV4D) and in NHBEs (NSUN2, NOP2, and CMTR2; Fig EV4E). This expression pattern may reflect host regulatory processes to compensate for the loss in activity of MTases, broadly inhibited by the activity of DZNep.

In order to explore the cellular functions perturbed by DZNep in NHBEs, we analyzed pathways associated with proteins, significantly regulated by DZNep in SARS-CoV-2- and SARS-CoV-infected conditions. We applied a network diffusion approach, which allows to highlight clusters of functionally related host proteins and pathways, which may be implicated in DZNep-induced perturbations (Wu *et al*, 2014). Among the significantly enriched subnetworks was a cluster of genes functionally interacting with STAT3 and NF-KB1 (Fig EV4F and G). In particular, this cluster can be subdivided into two distinct parts containing proteins related to biological processes governing fibrosis and blood coagulation, and inflammation (Fig EV4F). In line with these findings, we observed that DZNep treatment led to a reduction in pulmonary fibrosis biomarkers (e.g., COL4A1, MMP14, and SERPINE1) and upregulation of factors counteracting fibrotic processes (e.g., ELAFIN, SLPI, and ECM1; Fig 4D). Furthermore, it led to reduction in factors of the extrinsic coagulation cascade (e.g., F3 and TFPI2) and plasminogen activation system (e.g., PAI1, PLAT, PLAU), which were upregulated by SARS-CoV-2 (O'Sullivan *et al*, 2020; Jha *et al*, 2021; FitzGerald *et al*, 2021; Fig 4D). We also observed DZNep-dependent changes in

                                                                    

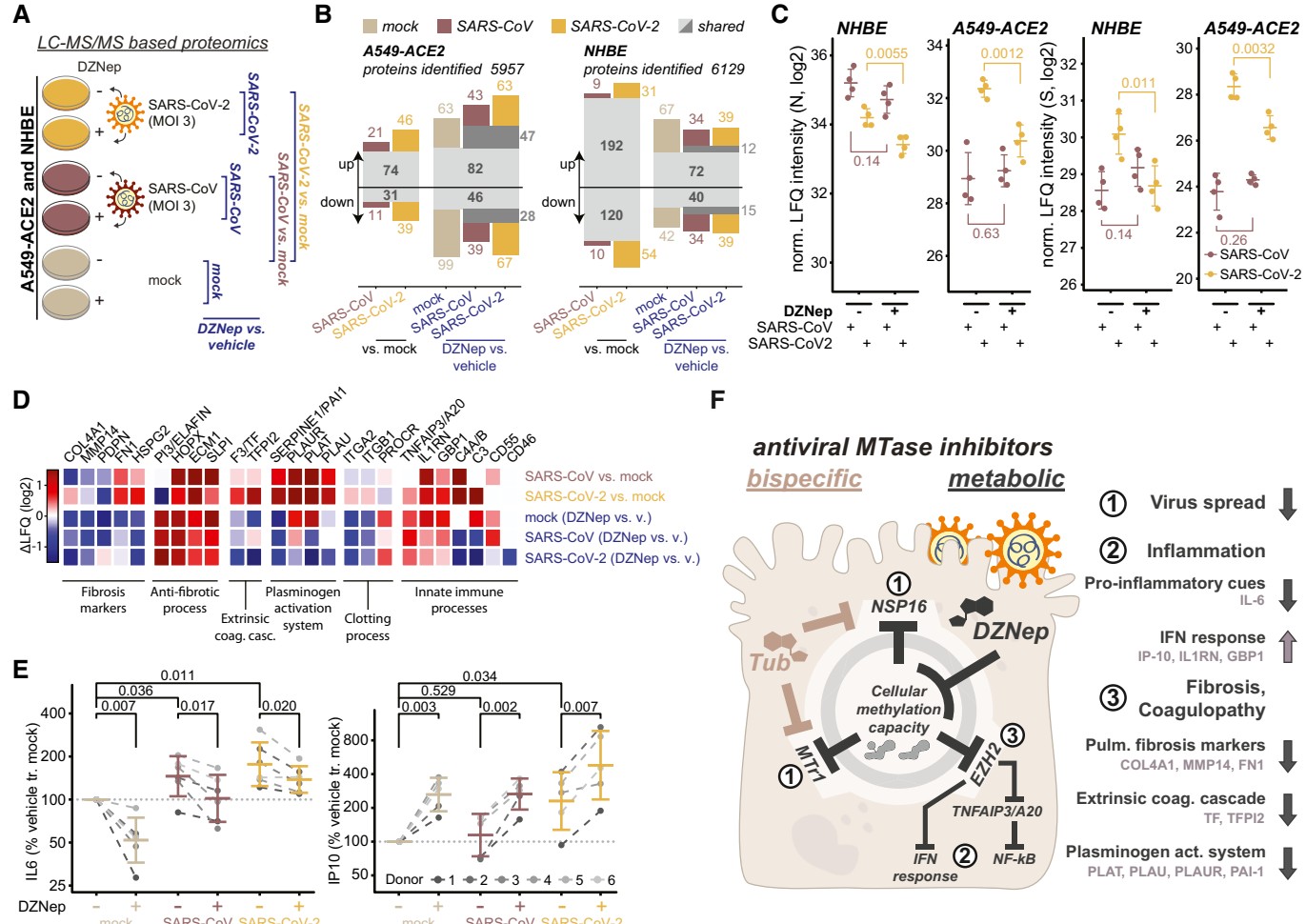

**Figure 4. DZNep treatment modulates tissue and immune processes.**

A–D  Mass spectrometry-based analysis of cells treated with DZNep and infected with SARS-CoV-2 and SARS-CoV. (A) Schematic representation of LC–MS/MS experiments. A549-ACE2s and NHBEs were pretreated for 6 h with 0.75 and 1.5 μM DZNep, respectively, or vehicle (PBS), and infected with SARS-CoV-2 or SARS-CoV at MOI 3 for 24 h (A549-ACE2) or 36 h (NHBEs). Changes in protein abundance were analyzed according to the depicted scheme using LASSO-based linear model followed by fixed LASSO inference-based p-value estimation as described in the Materials and Methods section. (B) Number of significantly up- or downregulated proteins in indicated comparisons according to (A). (C) Donor-normalized LFQ abundance of viral nucleoprotein (N) and spike (S) in the indicated conditions. Error bars represent mean ± SD of n = 4 donors (NHBE) or n = 4 independently infected A549-ACE2 cultures. Statistics were calculated using Student's two-sided t-test as indicated. (D) Expression patterns according to (A) of a selection of genes related to the disease-relevant pathways as annotated.

E  NHBEs (six independent donors) were pretreated for 6 h with 0.75 μM DZNep or vehicle and infected with SARS-CoV or SARS-CoV-2 at MOI 3. Twenty-four hours later, accumulation of IL-6 and IP-10 was measured in the supernatant by ELISA. Donor-wise IL-6 and IP-10 secretion, normalized to vehicle-treated uninfected controls (as further described in Materials and Methods), is shown.

F  Schematic representation of the proposed disease-relevant functions of DZNep in the context of COVID-19 alongside the model of their molecular origin.

abundance of innate immunity-related factors (e.g., IL-1RN, C3, and TNFAIP3/A20; Fig 4D). In particular, TNFAIP3/A20 was previously shown to be upregulated by DZNep, leading to taming of NF-kB signaling (Loong, 2013).

These findings prompted us to explore the impact of DZNep treatment on cell-intrinsic immunity ex vivo. Of particular relevance for infection-associated pathology are the overshooting pro-inflammatory cytokine secretion (i.e., cytokine storm; Blanco-Melo et al, 2020; Leisman et al, 2020) and blunted type I interferon signaling (Acharya et al, 2020; Hadjadj et al, 2020), which is inhibited by SARS-CoV-2 through multiple mechanisms (Miorin et al, 2020; Stukalov et al, 2021). Toward this, we quantified secretion of the

IRF3-dependent cytokine IP-10 and the NF-kB-dependent cytokine IL-6 by ELISA. As expected and reported previously (Leisman et al, 2020; Blanco-Melo et al, 2020), we observed a SARS-CoV and SARS-CoV-2 infection-dependent increase in IL-6 secretion in NHBEs (Fig 4E). Interestingly, DZNep treatment significantly reduced IL-6 secretion in all tested conditions (Fig 4E), which may be explained by its upregulation of TNFAIP3/A20 (Loong, 2013; Yang et al, 2020). In contrast to IL-6 and in line with MS-based observations concerning interferon-induced proteins (e.g., upregulation of IL-1RN and GBP1), IP-10 secretion was enhanced after DZNep treatment (Fig 4E). Collectively, these analyses indicate that DZNep treatment of SARS-CoV-2-infected primary human NHBEs not only inhibits

virus proliferation but also elicits favorable immunomodulatory and antifibrotic effects (Fig 4F). Notably, the combination of multiple beneficial activities could provide the required synergy for effective treatment of COVID-19 and its symptoms.

## SCIs treatment does not select for escape mutants in viral methyltransferases

Plasticity of SARS-CoV-2 genome was previously demonstrated *in vitro* upon treatment with remdesivir (Szemiel *et al*, 2021). SCIs include inhibitors of both SAM biosynthesis and SAH hydrolysis—while both perturb biomarkers of cellular methylation capacity, the former act by limiting SAM (Zhang *et al*, 2013) and the latter act by increasing SAH amounts (Aury-Landas *et al*, 2019). The two types of SAM cycle inhibition could thereby exert distinct selection pressures on the virus and in particular on viral MTases. In order to study how the virus may adapt to the SCI-induced metabolic reprogramming, we propagated SARS-CoV-2 in the presence of either DMSO (control), FIDAS-5 (2.5 μM), or DZNep (1.25 μM) for 10 passages (Figs 5A and EV5A). We observed a consistent reduction in SARS-CoV-2 titer upon treatment with DZNep and FIDAS-5 at early passages (Fig EV5A). To monitor the potential gradual adaptation of the virus to DZNep and pinpoint potentially affected viral proteins, we sequenced virus isolates at every passage. This analysis revealed acquisition of mutations that are associated with adaptation to cell culture conditions (e.g., Spike R685H; Sasaki *et al*, 2021) and an overall comparable number of mutations in all conditions, suggesting that the SCIs do not affect overall viral mutation rates (Fig EV5B–D and Dataset EV5). While we identified substitutions that correlated with increased virus titer upon cultivation (Fig EV5E), we did not observe mutations in the viral proteins associated with methylation processes (i.e., NSP10, NSP14, and NSP16). To directly compare the fitness of individual virus isolates, we performed virus competition experiments in the presence of SCIs using parental (P0) and passage 10 (P10) isolates (Fig 5A). DMSO-adapted control viruses and viruses propagated in the presence of DZNep (Fig 5B) or FIDAS-5 (Fig EV5F) grew similarly under most tested conditions, suggesting no adaptation of viruses propagated in the presence of SCIs (Fig EV5G and Dataset EV6). In contrast, the growth of DMSO-adapted control virus outcompeted growth of the P0 isolate (Figs 5B and EV5F), which likely reflects the adaptation of SARS-CoV-2 to *in vitro* cultivation. The lack of adaptive mutations in viral MTases and the lack of increased fitness upon propagation of SARS-CoV-2 in the presence of SCIs underline the challenge for SARS-CoV-2 to adapt to SCI treatments. These findings further support the suitability of host-directed SCIs to impair virus growth for therapeutic purposes.

## Synergistic potential of DZNep and its antiviral activity in a murine infection model

We next evaluated antiviral efficacy of DZNep in co-treatment with currently known COVID-19 drug candidates. In particular, we used dexamethasone (Carvalho *et al*, 2021; immunomodulatory corticosteroid), chloroquine (Carvalho *et al*, 2021; inhibitor of autophagy), ipatasertib (Stukalov *et al*, 2021; AKT kinase inhibitor), marimastat and prinomastat (Stukalov *et al*, 2021; hydroxamic acid-based broad-spectrum matrix metalloprotease inhibitors), remdesivir

(Carvalho *et al*, 2021; an antiviral nucleoside analog), and IFN-α. Toward this, we pretreated A549-ACE2 cells with DZNep and known antiviral compounds and monitored growth of the SARS-CoV-2-GFP. Under the tested conditions, DZNep did not impair, and was not impaired by, any of the tested drugs (Appendix Fig S1A–D). Cap 2′O-ribose methylation is often required by the viruses, including coronaviruses (Menachery *et al*, 2014, 2017), to evade cell-intrinsic immunity, specifically from being sensed by the cellular pattern recognition receptors RIG-I (Schuberth-Wagner *et al*, 2015) and MDA5 (Züst *et al*, 2011) and restricted by the IFN-induced protein IFIT1 (Daffis *et al*, 2010; Habjan *et al*, 2013; Abbas *et al*, 2017). Insufficiency in cap 2′O-ribose methylation of viral or host RNA could thereby promote and potentiate cell-intrinsic antiviral mechanisms to further restrict virus replication. In line with this hypothesis, we demonstrate synergistic potential between antiviral activities of DZNep and IFN-α *in vitro* (Figs 5C and EV5H). In coronavirus infections, SAM facilitates the association of viral MTase NSP16 with its allosteric activator NSP10 (Aouadi *et al*, 2017). Interestingly, besides with NSP16, NSP10 also interacts with NSP14 through an overlapping interface to greatly stimulate its ExoN (Bouvet *et al*, 2014; Ma *et al*, 2015) but not MTase activity (Bouvet *et al*, 2010, 2012). It is possible that the interaction between NSP14 and NSP10 is in a similar manner facilitated by SAM binding. SCIs could, in this respect, affect the resistance of SARS-CoV-2 to incorporable nucleoside analogs such as Remdesivir, activity of which is reduced by 4.5-fold through ExoN activity of NSP14 (Shannon *et al*, 2020). Notably, we demonstrate synergistic functions between DZNep and remdesivir *in vitro* (Figs 5D and EV5I). While the molecular mechanism behind these observations is yet to be explored, they suggest that modulating SAM cycle metabolite levels by SCIs may influence the ExoN activity of NSP14 of SARS-CoV-2.

DZNep was studied as an antitumor drug, and in rodents, it exhibits favorable pharmacokinetics for treating acute pulmonary infections (Bray *et al*, 2000; Peer *et al*, 2013; Sun *et al*, 2015). It has also been shown to support tissue regeneration (Xiao *et al*, 2016; Zeybel *et al*, 2017; Mimura *et al*, 2018), which is essential to mitigate virus-associated long-term complications. In order to test whether DZNep treatment is antiviral against SARS-CoV-2 *in vivo*, we infected C57BL/6 mice with SARS-CoV-2 beta variant (B.1.351, 250 pfu intranasal) and treated the animals at the day (day 0) and 1 day after infection (day 1) with DZNep (10 μg, intranasal application; Fig 5E). At day 2 post-infection, which represented the early acute stage of infection, the animals were sacrificed and lungs were harvested to quantify the viral load. We observed a significant reduction in the infectious viral load (Fig 5F), as well as diminished abundance of virus-derived mRNAs (Figs 5G and EV5J) in the lungs of DZNep-treated animals relative to the controls in the absence of any indication of toxicity (Fig 5H).

Taken together, we discovered a surprising relationship between SARS-CoV-2 NSP16 and cellular MTr1, which influences considerations on therapeutic approaches against COVID-19. We show that broad targeting of MTases involved in the viral life cycle by host-directed antivirals may be favorable over highly specific directly acting antivirals. Moreover, we show that the multispecific and metabolic MTase inhibitors, such as DZNep, are yet unexplored treatment options against COVID-19 (Fig 5I).

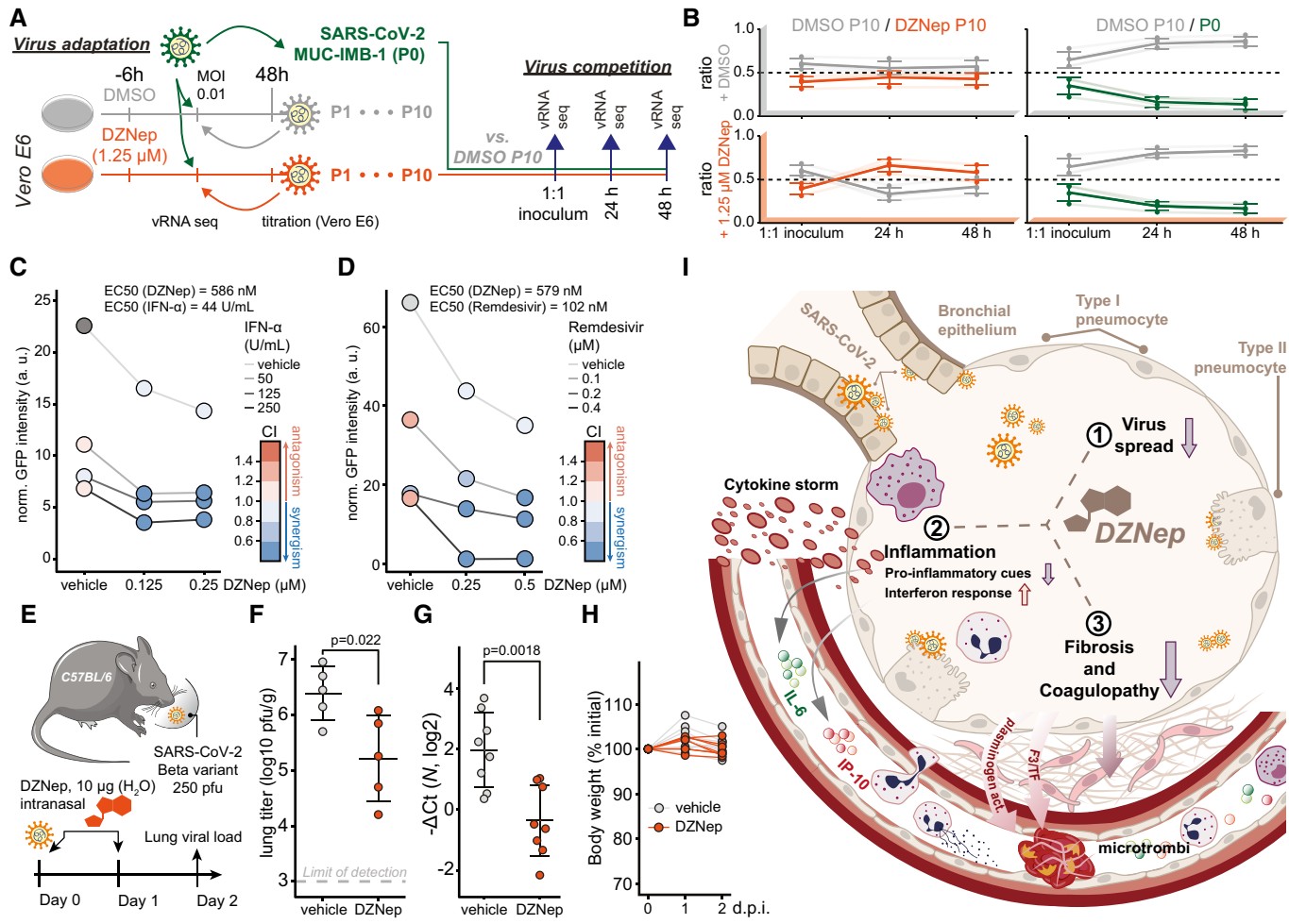

**Figure 5.  DZNep treatment does not lead to virus adaptation, exhibits synergism with remdesivir and IFN-α, and is antiviral *in vivo*.**

A    Schematic representation of the virus adaptation and subsequent pairwise competition experiments employing control (DMSO) and DZNep treatments.

B    Ratio of indicated pairs of viral isolates in 1:1 inocula and 24 and 48 h post-infection of Vero E6 cells undergoing treatments as annotated. Four individual variations were used for ratio calculation (Fig EV5G, full list is available in Dataset EV6) and are shown alongside means ± SD.

C    A549-ACE2 cells were pretreated for 6 h with indicated concentrations of IFN-α and DZNep and infected with SARS-CoV-2-GFP at MOI 1. Means of normalized integrated GFP intensities of six independently infected wells are shown as a measure of the reporter virus growth at 24 h post-infection alongside the combination index (CI) (Chou & Talalay, 1984) as a measure of treatments' synergy.

D    A549-nRFP-ACE2 cells were pretreated for 6 h with indicated concentrations of remdesivir and DZNep and infected with SARS-CoV-2-GFP at MOI 1. Means of normalized integrated GFP intensities of five independently infected wells are shown as a measure of the reporter virus growth at 24 h post-infection alongside the combination index (CI) (Chou & Talalay, 1984) as a measure of treatments' synergy. The presented data are representative of three independent repeats.

E    Schematic representation of the *in vivo* antiviral assay employing a murine infection model.

F, G    C57BL/6 mice were infected with SARS-CoV-2 beta variant (250 pfu, intranasal) and treated at D0 and D1 with DZNep (10 µg, intranasal). Forty-eight hours post-infection, lungs of infected mice were isolated. (F) Lung infectious viral load was quantified by titration of lung homogenate supernatants on Vero E6 cells and expressed as log10 plaque-forming units per unit mass alongside mean ± SD (*n* = 5 animals per condition). Statistics were calculated using Student's two-sided *t*-test as indicated. (G) Abundance of viral transcript encoding SARS-CoV-2 N was quantified in the lung samples by RT–qPCR. The graph shows negative ΔCt values, as normalized to *18S rRNA*, and respective mean ± SD (*n* = 8 animals per condition). Statistics were calculated using Student's two-sided *t*-test as indicated.

H    Animal body weight measured at indicated times post-infection, depicted as percentage of initial weight.

I    Schematic representation of the proposed disease-relevant functions of DZNep in the context of COVID-19.

Data information: Data shown in (G, H) were pooled from two independent experiments.
Source data are available online for this figure.

## Discussion

Here, we showed that robust antiviral effect against SARS-CoV-2 via cap 2'-O-ribose MTase inhibition requires concomitant targeting of both host (MTr1) and viral (NSP16) MTases. While the molecular mechanism behind the MTr1 involvement in the viral life cycle is yet to be clearly delineated, given its analogous function in host mRNA maturation, it is likely that it adds a degree of redundancy to the NSP16-facilitated viral RNA methylation. One may question why SARS-CoV-2 is not entirely relying on the activity of MTr1 given space constraints in viral genomes. A possible explanation may be the suboptimal localization of MTr1 from the viral perspective.

Alternatively, the readily available methylation capacity conferred by MTr1 may not be sufficient to facilitate rapid viral RNA buildup in the early stage of infection. Moreover, expression of MTr1 is induced by IFNs (Bélanger et al, 2010) but IFN expression is heavily inhibited by SARS-CoV-2. Collectively, the surprising synergy observed between NSP16 and MTr1 indicates that methylation of viral RNA is a rate-limiting step in the viral life cycle that cooperatively leverages the activities of both cellular and viral factors. The engagement of MTr1 seems to be specific for SARS-CoV-2 and has, to our knowledge, not been reported for any other virus that employs de novo RNA maturation. Indeed, cap 2'O-ribose methyltransferase activity-deficient YFV replicated similarly in MTr1-deficient cells, suggesting that the cellular RNA methyltransferase is not promiscuously active on viral RNAs. This novel conceptual advancement has direct implications for antiviral drug design and facilitated the discovery of two novel classes of antivirals active against SARS-CoV-2 that synergistically influence both host and viral factors.

Using in silico docking validated by in vitro MTase activity assays, we show that tubercidin is a broad-spectrum MTase inhibitor active against both NSP16 and MTr1. Tubercidin has been studied for antiviral (Olsen et al, 2004; Vittori et al, 2006) and anticancer properties (Grage et al, 1970), but its therapeutic value was hampered by unfavorable in vivo pharmacologic properties. As an alternative approach, we envisioned to target the SAM cycle through SCIs, which would indirectly hamper the activities of both NSP16 and MTr1. In contrast to tubercidin, the SCI DZNep is a well-tolerated drug that competitively inhibits AHCY at picomolar concentrations (Glazer et al, 1986) and that was also studied as an anticancer drug (Bray et al, 2000; Peer et al, 2013; Sun et al, 2015). In rats, SCI DZNep exhibits a favorable lung-to-plasma ratio of 3, its no-observed-adverse-effects level (NOAEL) was 10 mg/kg (Sun et al, 2015), and it is rapidly eliminated through renal secretion (Peer et al, 2013; Sun et al, 2015). Despite this, a single dose of DZNep was highly efficacious against Ebola (Bray et al, 2000, 2002) and vesicular stomatitis virus infections (De Clercq et al, 1989) in mice. In rats, liposome packaging of DZNep was shown to increase the area under the plasma concentration curve by 138-fold (Sun et al, 2012), potentially offering a major reduction in treatment doses. DZNep was previously shown to have antiviral activity against some viruses but not against others (Tseng et al, 1989; Chen et al, 2013; Arbuckle et al, 2017). In particular, it was shown to be antiviral against Ebola virus in mice (Bray et al, 2000, 2002), which could be explained by a combination of interferon induction and impeded viral RNA maturation. DZNep was also shown to impair viral mRNA cap methylation in the context of vesicular stomatitis virus infection and reduce viral mRNA translation (Gibbons et al, 2021). Recently, DZNep was reported to be antiviral against SARS-CoV-2 in vitro and in ovo (Kumar et al, 2022), leading to viral RNA m6A methylation and cap maturation defects and consequently to reduced viral protein production and inhibition of virus replication (Kumar et al, 2022). Overall, the above-mentioned independent work offers further evidence and supports findings presented herein toward demonstrating the treatment potential of SCIs against COVID-19.

The broad activity of DZNep on individual potentially antiviral pathways (IFNs, host or viral RNA methylation, histone methylation, etc.) complicates the identification of a dominant antiviral mechanism. Disregarding the potential contribution of inhibiting NSP14, the concomitant inhibition of NSP16 and MTr1 through drug-induced SAM starvation and SAH-based product inhibition may on its own restrict SARS-CoV-2 proliferation. Supportive of this hypothesis, it was previously shown that SAM facilitates the association of MERS-CoV NSP16 with its allosteric activator NSP10, and that SAH inhibits the MTase activity of NSP10/16 in vitro (Aouadi et al, 2017). Activity of NSP16 was also shown to be required for IFN resistance and virulence of related SARS and MERS coronaviruses (Menachery et al, 2014, 2017). Hypomethylation of viral mRNA at cap-proximal ribose moieties could promote antiviral innate immune activation (Züst et al, 2011; Schuberth-Wagner et al, 2015) and sensitize the virus towards translational repression by the innate immune effector protein IFIT1 (Daffis et al, 2010; Habjan et al, 2013; Abbas et al, 2017). Collectively, these effects could in part explain the DZNep-induced amplification of antiviral signaling that we observed ex vivo (Figs 4D and E, and EV4E and F), and synergism of DZNep with IFN-α co-treatment observed in vitro (Fig 5C), which induces expression of IFIT1, as well as shed light on selective antiviral efficacy of DZNep against SARS-CoV-2 as opposed to less IFN-sensitive SARS-CoV (Lokugamage et al, 2020). However, DZNep was active in a STAT1-independent manner (Fig EV3A and H), suggesting that innate immune signaling only partially contributes to anti-SARS-CoV-2 activity of SCIs.

Beyond suppressing virus growth, COVID-19 has numerous pulmonary and extrapulmonary manifestations requiring separate pharmaceutical interventions (Gupta et al, 2020). Coagulopathy, characterized by elevated von Willebrand factor, fibrinogen, and D-dimers and leading to excessive thrombin production, inhibition of fibrinolysis, and complement activation, has been associated with infection-mediated endothelialitis and endothelial injury (Gupta et al, 2020; Varga et al, 2020). We show that in primary human lung cells, SARS-CoV-2 and to a lesser extent SARS-CoV infections influenced abundance of proteins involved in blood coagulation such as PLAT (t-PA), PLAU (u-PA), PLAUR (u-PAR), F3 (TF), and SERPINE1 (PAI1), as well as components of the complement system such as C3, C4A/B, CD46, and CD55 (Fig 4D). Notably, DZNep treatment reduced the infection-induced deregulation of the above-mentioned factors. In most cases, the effect of DZNep treatment was also observed in uninfected settings, suggesting that this activity is unrelated to repression of virus growth and thus depends on inhibition of a yet unidentified host MTase. Crosstalk between fibrinolysis and organ fibrosis is mediated through protease antiprotease balance that further dictates tissue remodeling and cytokine activation (Mercer & Chambers, 2013). Using primary human lung cells, we show that SARS-CoV-2, and to a lesser degree SARS-CoV, perturbs abundance of pulmonary fibrosis-related proteins such as SERPINE1/PAI1, FN1, and HSPG2 (Fig 4D). Furthermore, we show that DZNep treatment alone or in context of SARS-CoV or SARS-CoV-2 infections reduces abundance of pulmonary fibrosis markers (e.g., SERPINE1, MMP14, and COL4A1) and increases levels of factors with antifibrotic activity (e.g., HOPX, PI3/ELAFIN, and SLPI; Fig 4D). These observations are in line with previous reports describing antifibrotic activity of DZNep in lungs (Xiao et al, 2016), liver (Zeybel et al, 2017), and kidneys (Mimura et al, 2018), which was linked to drug-induced inhibition of EZH2. Similar modulation of fibrosis-related proteins may be induced by other SCIs beyond DZNep, which may also perturb EZH2 activity in a metabolite-mediated manner.

The hallmark immunological characteristic of severe COVID-19 is the cytokine imbalance, whereby strong pro-inflammatory cues (e.g., mediated by elevated IL-6) are accompanied by only minor activation of innate antiviral defenses (e.g., mediated by type I IFNs) leading to deleterious systemic response (Blanco-Melo *et al*, 2020). It was shown that the type I IFN response is highly antiviral against SARS-CoV-2 (Mantlo *et al*, 2020). For this reason, interferons were considered as therapeutic options for COVID-19, but are as of yet not clinically used (Alavi Darazam *et al*, 2021). On the contrary, suppression of overshooting pro-inflammatory cues by, e.g., dexamethasone limits inflammation-mediated lung injury and is widely used for the treatment of COVID-19 (The RECOVERY Collaborative Group, 2021). Numerous biologicals (e.g., anti-IL-6 receptor (The REMAP-CAP Investigators, 2021) or anti-GM-CSF (De Luca *et al*, 2020)) were proposed to be used in a similar manner. We show that DZNep treatment is sufficient to prime and boost the cell-intrinsic antiviral response *ex vivo*, evidenced by upregulation of immunity-related genes (e.g., GBP1, IL-1RN; Fig 4D) and cytokines (IP-10; Fig 4E) in both uninfected and infected settings. In contrast, DZNep treatment of primary human lung cells led to upregulation of TNFAIP3/A20 (Fig 4D) and reduced SARS-CoV-2-induced secretion of NF-kB-dependent cytokine IL-6 (Fig 4F). In line with our observations, DZNep treatment was previously shown to inhibit EZH2 and invoke depletion of H3K27 trimethylation (Tan *et al*, 2007; Miranda *et al*, 2009), lifting the epigenetic suppression of the antiviral interferon signaling (Bray *et al*, 2002; Wee *et al*, 2014; Tiffen *et al*, 2020), as well as leading to upregulation of the NF-kB inhibitor TNFAIP3/A20 (Loong, 2013) and subsequent inhibition of NF-kB signaling (Loong, 2013). Specific inhibitors of EZH2, such as tazemetostat, may thus be effective co-treatment options along antivirals used in treatment of COVID-19 and other infectious diseases. The critical reliance on viral or host MTases and labile nature against intrinsic antiviral responses are common themes across the spectrum of pathogenic viral families. Supported by previous reports of its antiviral efficacy against diverse viral pathogens, the combined activities of DZNep and potentially SCIs in general make them unique candidate broad-spectrum antivirals that could curb multiple aspects of disease progression beyond COVID-19.

Herein, we leveraged both direct-acting and host-directed antiviral drug repurposing to explore the antiviral potential of pharmaceutical inhibition of SARS-CoV-2 cap 2'-O-ribose methyltransferase NSP16. We demonstrate that robust antiviral effect against SARS-CoV-2 critically requires concomitant inhibition of both viral MTase NSP16 and a novel host factor MTr1. Collectively, presented findings emphasize the potential of developing cross-functional host-directed antivirals, wherein the state-of-the-art knowledge of both virus and host biology is leveraged for applied antiviral research. Notably, we showed that host-directed therapies acting on the virus–host metabolic interface and targeting the SAM cycle can possess both antiviral and host-preserving functions. To our knowledge, no single-drug therapies against COVID-19 that would also tackle potentially long-term lung damage and fibrosis are currently available (Chitalia & Munawar, 2020; George *et al*, 2020; Feuillet *et al*, 2021). Most notably, DZNep is unparalleled in combining these activities by repressing viral load, limiting the hyperinflammatory response and promoting cell-intrinsic tissue repair programs, making it and similarly acting SCIs excellent treatment candidates against COVID-19.

## Materials and Methods

### Cell lines and reagents

HEK293T, A549 (kindly provided by Takashi Fujita, Kyoto University, Kyoto, Japan), A549-ACE2, and Vero E6 (CRL-1586, ATCC) cell lines, and their respective culturing conditions, were described previously (Stukalov *et al*, 2021). NHBE cells (CC-2540, Thermo Fisher Scientific) were cultured as described previously (Zissler *et al*, 2016); in short, the cells were grown until reaching 80% confluence. To avoid gene expression changes or influence on virus growth induced by growth factors in the BEGM (Lonza), cells were rested in basal medium (BEBM, Lonza) for 24 h before the start of the experiment. Baby hamster kidney cells (BHK-21/J), kindly provided by Charles M. Rice, Rockefeller University, New York, NY, USA, were grown in MEM containing 7.5% FBS, 1% L-glutamine, and 1% non-essential amino acids at 37°C, with 5% $CO_2$. Calu-3 cells (kindly provided by Stephan Pöhlmann, Deutsches Primatenzentrum, Münster, Germany, and Stephan Ludwig, University of Münster, Münster, Germany) were maintained in Eagle's Minimum Essential Medium (MEM), containing 1% non-essential amino acids (NEAA, Gibco 11140), 10% fetal bovine serum, 1 mM Na-pyruvate (Gibco, 11360), 100 U/ml penicillin, and 100 µg/ml streptomycin at 37°C, with 5% $CO_2$. A549-RFP-ACE2 cell line was generated through lentiviral transduction of A549-ACE2 cell line and blasticidin selection, leading to expression of nuclear localized mRFP—plasmid pHIV-H2BmRFP was a gift from Bryan Welm & Zena Werb (Addgene plasmid #18982; http://n2t.net/addgene:18982; RRID: Addgene_18982; Welm *et al*, 2008). In preparation of KO cell lines, the following sequences were used in a multiplexed manner for cloning of gRNA templates into pLentiCRISPRv2 plasmid: STAT1 (GGTGGCAAATGAAACATCAT; GAGGTCATGAAAACGGATGG; CAG GAGGTCATGAAAACGGA), NTC (Sanjana *et al*, 2014; AACCGGAT CGCCACGCGTCC; TCCGGAGCTTCTCCAGTCAA; TGCAAAGTTCA GGGTAATGG), AHCY (TTTCCTCCCGTAGCCGACAT; CCAGGCAGCC AGGCCGATGT; TCCCGTAGCCGACATCGGCC) and MAT2A (CTGG AATGATCCTTCTTGCT; TGGAATGATCCTTCTTGCTG; TGCTGTT GACTACCAGAAAG). pLentiCRISPRv2 was a gift from Feng Zhang (Addgene plasmid #52961; http://n2t.net/addgene:52961; RRID: Addgene_52961; Sanjana *et al*, 2014). Lentivirus production, transduction of cells, and antibiotic selection for KO preparation were performed as described previously (Stukalov *et al*, 2021). In brief, A549-ACE2 cells were transduced using puromycin resistance carrying lentiviruses encoding Cas9 and gRNAs and grown for 4 days using medium, supplemented with 3 µg/ml puromycin, before being used for further experiments. MTr1 KO cells (clone number: H1) were generated by transducing the parental A549 cells with plasmid encoding gRNA (CCTCAACGATGTCCTTCCGACCC), and Cas9 and mCherry (kindly provided by Martin Schlee). After FACS sorting for mCherry-positive cells, clonal colonies were isolated, expanded, and validated for the loss of MTr1 by Western blotting and genome sequencing (CTR cell line was selected from clones with intact MTr1 locus and expression). All cell lines were tested to be mycoplasma-free.

For the stimulation of cells, recombinant human IFN-α was a kind gift from Peter Stäheli. The following inhibitors were used: SAH (RayBiotech, 229–20003), sinefungin (Cayman Chemical Company, 13829), cladribine (Cayman Chemical Company, Cay12085-

50), clofarabine (Cayman Chemical Company, B2764-Cay14125-10), 2-fluoroadenosine (Sigma-Aldrich, 656402), fludarabine (Tocris, 3495), 2-aminoadenosine (Santa Cruz Biotechnology, sc-220693A), vidarabine (BLD Pharmatech, BD42581), 3-DZA (Cayman Chemical Company, 9000785), nebularine (Cayman Chemical Company, 31329), tubercidin (Sigma-Aldrich, T0642), 1-DZA (Tocris, 4488), ribavirin (Sigma-Aldrich, R9644), tecadenoson (BLD Pharmatech, BD00781750), ITU (Sigma-Aldrich, I100), GDP-D-mannose disodium salt (Sigma-Aldrich, 07508), 3-deazaneplanocin A (Sigma-Aldrich, 5060690001, and Biozol, SEL-S7120), D-eritadenine (Biomol, Cay21747-1), remdesivir (Hölzel Biotech, CS-0028115), FIDAS-5 (MAT2A Inhibitor II, FIDAS-5—Calbiochem, Sigma-Aldrich, 5041730001), MAT2A inhibitor 1 (Hölzel Diagnostika, HY-112131), PF-9366 (Hölzel Diagnostika, HY-107778), CBHcy (S-(4-Carboxybutyl)-D,L-homocysteine, BioTrend, AOB2142), tazemetostat (EPZ-6438, Biomol, Cay16174-1), dexamethasone (Sigma-Aldrich, D1756), marimastat (Sigma-Aldrich, M2699), prinomastat (Sigma-Aldrich, PZ0198), ipatasertib (GDC-0068, 18412, Cayman chemical), and chloroquine (Chloroquine diphosphate salt, Sigma-Aldrich, C6628).

For the detection of protein abundance by Western blotting, ACTB-HRP (Santa Cruz; sc-47778; 1:5,000 dilution), ACE2 (Abcam; ab15348; 1:1,000 dilution), Venus (Santa Cruz; sc-9996; 1:1,000 dilution), MTr1 (Novus bio; NBP1-83047; 1:1,000 dilution), hnRNPA1-HRP (Santa Cruz; sc-32301 HRP; 1:1,000 dilution), GAPDH-HRP (Cell Signaling; 3683S; 1:1,000 dilution), and SARS-CoV-2/SARS-CoV N protein (Sino Biological; 40143-MM05; 1:1,000 dilution) antibodies were used. Secondary antibody detecting mouse IgG (Cell Signaling; 7076; 1:5,000 dilution) was horseradish peroxidase (HRP)-coupled. Alexa Fluor 488-conjugated goat anti-mouse antibody (Abcam, ab150113) was used for protein abundance detection by immunofluorescence. WB imaging was performed as described previously (Stukalov *et al*, 2021).

### Structure-based NSP16 inhibitor screening

Structure-based virtual screening for NSP16 inhibitors was conducted using molecular docking against 5,597 bioactive compounds, with molecular weights ranging from 200 to 800 Da, from the Drug-Bank database. Docking simulations were performed using the Glide (Friesner *et al*, 2004; Halgren *et al*, 2004) SP docking program (Schrödinger, LLC) with a grid box defined by the SAM-binding pocket from the crystal structure of SARS-CoV-2 NSP10/16 (PDB ID: 6W4H).

UMAP dimensionality reduction according to MACCS structural keys (chemicalchecker.org; Duran-Frigola *et al*, 2020) was performed in Python 3.8.5, package UMAP 0.5.1, using default parameters.

### Methyltransferase assays

Cap0 RNA, the methyl group acceptor in methyltransferase assays, was synthesized using the HiScribe T7 Quick High Yield RNA Synthesis Kit (NEB, E2050S) with cap analog $m^7G$ (5′)ppp (5′)A (NEB, S1405), according to the manufacturer's instructions. The annealed 5′-overhang dsDNA was used as a template (Sense: 5′-TAATAC GACTCACTATA-3′, Antisense: 5′-CACTTTCACTTCTCCCTTTCAG TTTCCCTATAGTGAGTCGTATTA-3′).

The reaction buffer (50 mM Tris–HCl (pH 8.0), 5 mM KCl, 1 mM $MgCl_2$, 1 mM DTT) was complemented with methyltransferases (5 U/μl VACV VP39 (NEB, M0366S) or 1.5 μM/0.7 μM SARS-CoV-2 Nsp10/16 (Biomol, BPS-100747-1)), 10 mM tubercidin (or DMSO as vehicle control), 17 μM $m^7GpppApG$ ($pN_{27}$; cap0 RNA), and 1.2 μM (0.02 μCi/μl) SAM[$^3$H] (PerkinElmer, NET155V250UC). The reaction mixtures were incubated at 37°C overnight. The samples were purified using a mini Quick Oligo column (Roche, 11814397001) to remove free SAM[$^3$H]. The purified sample was diluted in ULTIMA GOLD (PerkinElmer, 6013329) and measured using a scintillation counter LS6500 (Beckman Coulter).

### Virus strains, stock preparation, and *in vitro* infection

SARS-CoV-Frankfurt-1 (Pfefferle *et al*, 2009), SARS-CoV-2-MUC-IMB-1 (Thi Nhu Thao *et al*, 2020), SARS-CoV-2 Alpha (B.1.1.7; Coronaviridae Study Group of the International Committee on Taxonomy of Viruses, 2020), SARS-CoV-2 Delta (B.1.617.2; Mlcochova *et al*, 2021), and SARS-CoV-2-GFP (Stukalov *et al*, 2021) strains were produced as described previously (Stukalov *et al*, 2021). The SARS-CoV-2 beta variant (B.1.351) was isolated in Bonn from a throat swab of a patient on and propagated on Caco-2 cells cultured in DMEM (10% FCS, 100 μg/ml streptomycin, 100 U/ml penicillin, and 2.5 μg/ml amphotericin B). All experiments with SARS-CoV-2 were performed in BSL3 laboratories under the approval of the Regierung von Oberbayern, Germany (AZ: 55.1GT-8791.GT_2-365-10 and 55.1GT-8791.GT_2-365-20) and approval of the government of Cologne, Germany. For *in vivo* experiments, the virus was passaged once on Caco-2 cells in DMEM (10% FCS, 100 μg/ml streptomycin, and 100 U/ml penicillin) at an MOI of 0.001 and harvested at 3 days post-infection. Virus in the cleared supernatant (200 g, 10 min, 4°C) was stored at −80°C. Viral titers of the stocks were determined on Vero E6 cells using a carboxymethylcellulose overlay as described previously (Koenig *et al*, 2021). Recombinant SARS-CoV-2 NSP16mut virus was generated via plasmid pBeloCoV harboring the inactivating mutations D130A and K170A in the coding sequence of NSP16 (pBeloCoV-NSP16mut), which was cloned through Red recombination (manuscript by T. Gramberg in preparation). The virus was further amplified in CaCo-2 cells (1 passage, 72 h) and quantified in cleared and purified supernatants by RT–qPCR. Recombinant YFV 17D and YFV 17D NS5 E218A (YFV E218A) were generated via electroporation of an infectious cDNA clone-derived *in vitro* mRNA transcript into BHK-J cells and a single passage on BHK-J cells; titers were determined by plaque assays using BHK-J cells as described previously (Kümmerer & Rice, 2002). Recombinant vesicular stomatitis virus (VSV) Indiana strain encoding EGFP in position 5 of the genome (VSV-GFP) was recovered from BSR T7/5 cells infected with VACV WR vTF7.3 and transfected with pVSV1 (+) P5_EGFP, pL, pP, and pN as described previously (Whelan *et al*, 1995). It was further propagated in BSR T7/5 cells, and virus titers were determined by the plaque assay using Vero cells.

Cells were pretreated with inhibitors by medium (containing any indicated inhibitor) exchange at 6 h (unless stated otherwise) prior to the addition of infectious inoculum containing SARS-CoV-2 at MOI 3 (SARS-CoV-2-MUC-IMB-1, unless stated otherwise) with medium replacement 1 h post-infection where indicated. Infection with YFV 17D wt and YFV NS5 E218A was performed in PBS

containing 1% FBS for 1 h, followed by 2× PBS and 1× MEM wash and medium replacement.

At the time of sample harvest, the cells were washed once with 1× PBS buffer and lysed in LBP (Macherey-Nagel), 1× SSB lysis buffer (62.5 mM Tris–HCl, pH 6.8; 2% SDS; 10% glycerol; 50 mM DTT; and 0.01% bromophenol blue), or freshly prepared SDC buffer (100 mM Tris–HCl, pH 8.5; 4% SDC) for RT–qPCR, Western blot, or LC–MS/MS analyses, respectively. The samples were heat-inactivated and frozen at −80°C until further processing. Sampled supernatants were stored frozen at −80°C until further processing.

### Antiviral assays using SARS-CoV-2-GFP

A549-ACE2 cells were seeded into 96-well plates in DMEM (10% FCS, 100 μg/ml streptomycin, 100 IU/ml penicillin) 1 day before infection. Six hours before infection, the medium was replaced with 125 μl of DMEM containing either the compound (s) of interest or their respective vehicle (s) as control. Infection was performed by adding 10 μl of SARS-CoV-2-GFP (MOI 3, unless otherwise stated) per well, and plates were placed in the IncuCyte S3 Live-Cell Analysis System where images of phase, green, and red (when using A549-RFP-ACE2 cell line) channels were captured at regular time intervals at 4× (whole-well) or 20× magnification. Cell viability was assessed as the cell confluence per well (phase area). Virus growth was assessed as GFP integrated intensity normalized to cell confluence per well (GFP integrated intensity/phase area) or GFP area normalized to cell confluence per well (GFP area/phase area) or GFP area normalized to RFP-positive nucleus count (when using A549-RFP-ACE2 cell line). Basic image analysis and image export were performed using the IncuCyte S3 software (Essen Bioscience; version 2019B Rev2). Statistical analysis and visualization were performed using R version 4.0.2. Three-parameter logistic function fitting was performed using R package drc (version 3.0-1).

### Plaque assays

Confluent monolayers of Vero E6 cells were infected with serial five-fold dilutions of virus supernatants (from 1:100 to 1:7,812,500) for 1 h at 37°C. The inoculum was removed and replaced with serum-free MEM (Gibco, Life Technologies) containing 0.5% carboxymethylcellulose (Sigma-Aldrich). Two days post-infection, cells were fixed for 20 min at room temperature with formaldehyde directly added to the medium to a final concentration of 5%. Fixed cells were washed extensively with PBS before staining with $H_2O$ containing 1% crystal violet and 10% ethanol for 20 min. After rinsing with PBS, the number of plaques was counted and the virus titer was calculated.

### Quantification of gene expression in cell lines by RT–qPCR

Total cellular RNA, or RNA content of the supernatants, was harvested and isolated using MACHEREY-NAGEL NucleoSpin RNA mini kit according to the manufacturer's instructions. Reverse transcription was performed using the Takara PrimeScript RT Reagent kit with gDNA eraser according to the manufacturer's instructions.

RT–qPCR was performed using primers targeting SARS-CoV-2 N (fw: 5′-TTACAAACATTGGCCGCAAA-3′; rev: 5′-GCGCGACATT CCGAAGAA-3′), SARS-CoV-2 *E* (Figs 1C and I, and EV1D; fw: 5′-AC AGGTACGTTAATAGTTAATAGCGT-3′; rev: 5′-ATATTGCAGCAGT ACGCACACA-3′), SARS-CoV-2 *E* (Fig EV1G) and MERS-CoV *N*, which were described previously (Matsuyama *et al*, 2020); SARS-CoV *N* as described previously (Corman *et al*, 2012); VSV N (fw: 5′-GGAGTATCGGATGCTTCCAGAACCA-3′; rev: 5′-ACGACCTTCTGGC ACAAGAGGTT-3′), MAT2A (fw: 5′-CTTCGTAAGGCCACTTCCGC-3′; rev: 5′-TCTGGTAGCAACAGCAGCTC-3′), AHCY (fw: 5′-AACTGC CCTACAAAGTCGCC-3′; rev: 5′-ATGGTCCTGGGTGGAGAAGA-3′), and RPLP0 (unless stated otherwise the housekeeper control, fw: 5′-GGATCTGCTGCATCTGCTTG-3′; rev: 5′-GCGACCTGGAAGTCCAA CTA-3′) using PowerUp SYBR Green (Thermo Fisher, A25778); and SARS-CoV-2 RdRp (fw: 5′-GTGAAATGGTCATGTGTGGCGG-3′; rev: 5′-CAAATGTTAAAAACACTATTAGCATA-3′; VIC-CAGGTGGAACC TCATCAGGAGATGC-BMN-Q535), Eukaryotic 18S rRNA (Hs99999901_s1, Applied Biosystems), human *IFNB1* (Hs01077958_s1, Applied Biosystems), human *IFIT1* (ISG56; Hs03027069_s1, Applied Biosystems), and human *MxA* (Hs00895608_m1, Applied Biosystems) using TaqMan Fast Advanced Master Mix (Applied Biosystems). QuantStudio 3 Real-Time PCR System (Thermo Fisher) or Step One Plus Real-Time PCR System (Applied Biosystems) was used. Ct values, obtained using QuantStudio Design and Analysis Software v1.4.3, were averaged across technical replicates and −ΔCt values as a measure of gene expression were calculated as Ct (RPLP0) − Ct (N). −ΔΔCt values as a measure of change in gene expression between distinct KOs and NTC were calculated as −ΔCt (KO) − (−ΔCt (NTC)). For display of highly divergent values, one replicate of vehicle-treated samples was assigned a relative expression value of $10^6$. Viral RNA copy was calculated from the standard curve using serial diluted cDNA with known copy number. Statistical analysis and visualization were performed using R version 4.0.2.

### Protein abundance quantification by Western blotting

At the time of sample harvest, the cells were washed with PBS and lysed in SSB buffer (62.5 mM Tris–HCl from 1 M stock solution with pH 6.8, 2% SDS, 10% glycerol, 50 mM DTT, and 0.01% Bromophenol Blue in distilled water), and protein concentrations were measured using Pierce 660-nm Protein Assay with an addition of Ionic Detergent Compatibility Kit (Thermo Fisher Scientific) according to the manufacturer's instructions. Protein concentrations were equalized, and up to 10 μg of proteins was loaded in NuPAGE Bis-Tris, 1 mm, 4–12% gels (Thermo Fisher Scientific). Protein separation was performed according to the gel manufacturer's instructions, and proteins were transferred to 0.22-μm nitrocellulose membrane (1 h at 100 V in 25 mM Trizma base, 0.192 M Glycine, pH 8.3). The membranes were blocked for 1 h in 5% skim milk in TBS-T buffer (0.25% Tween-20 in phosphate-buffered saline solution) with gentle agitation. The antibodies listed in the section above (cell lines and reagents) were diluted in 5% skim milk (TBS-T); the membranes were washed 5× for 5 min with TBS-T between and after incubations with primary and secondary antibodies. Western Lightning ECL Pro (PerkinElmer) was used for band detection according to the manufacturer's instructions. Normalization of band signals was performed using the Image Lab Software (Bio-Rad; version 6.0.1 build 34).

## MTr1 detection in separated cellular fraction

Cytoplasmic and nuclear extracts were prepared as described previously (Bélanger et al, 2010). Briefly, A549 cells with or without overnight IFN-β 1a (PBL Assay Science, 11410-2) treatment (1,000 U/ml) were detached from cell culture dish and resuspended in 600 µl of cold lysis buffer (10 mM HEPES (pH 7.5), 10 mM KCl, 0.1 mM EDTA, 0.1 mM EGTA, 1 mM DTT, and 1 × Protease/phosphatase Inhibitor Cocktail (Cell Signaling Technology, 5872S)). The cell suspension was incubated on ice for 15 min before addition of NP-40 to the final concentration of 0.5% and 10-s vortexing. The resulting mixture was centrifuged at $2,000 \times g$ for 30 s at 4°C before the supernatant (cytoplasmic fraction) was removed. The pellet was resuspended in 100 µl of nuclear extraction buffer (20 mM HEPES (pH7.5), 400 mM KCl, 1 mM DTT, and 1 × Protease/phosphatase Inhibitor Cocktail), incubated on a rotating wheel at 4°C for 15 min, and centrifuged at $14,000 \times g$ for 15 min at 4°C. The supernatant (nuclear fraction) was further harvested and frozen at −20°C until further use.

## Viral protein detection and quantification by immunofluorescence

For detection of viral protein expression using immunofluorescence, the cells were washed 3× with phosphate-buffered saline (PBS), fixed for 15 min with 4% formaldehyde in PBS, washed again, and permeabilized using 0.1% Triton-X in 4% BSA (PBS) for 15 min. They were further blocked for 1 h using 4% BSA in PBS. The antibodies listed in the section above (cell lines and reagents) were diluted in 4% BSA (PBS); the cells were washed 5× for 1 min with PBS between and after incubations with primary and secondary antibodies. Stained cells were imaged using IncuCyte S3 Live-Cell Analysis System. Whole-well images of GFP and Phase channels were captured at 4× magnification. Cell viability and virus growth were assessed as the cell confluence per well (phase area) and GFP integrated intensity normalized to cell confluence per well (GFP integrated intensity/phase area), respectively, using the IncuCyte S3 Software (Essen Bioscience; version 2019B Rev2). Analysis and visualization were performed using the R version 4.0.2.

## Quantification of secreted cytokines by ELISA

For detection of human IL-6 and IP-10, commercially available ELISA kits were used (Human IL-6 ELISA Set, BD OptEIA, 555220; Human IP-10 ELISA Set, BD OptEIA, 550926) according to the manufacturer's instructions. Basal medium, used for NHBE culturing at time of treatment and infection, was used as blank control. Statistics (Fig 4E) were calculated using paired Student's two-sided t-test on log-transformed values between indicated conditions before donor-wise normalization to vehicle-treated mock controls.

## Mass spectrometry sample preparation and analysis

For the determination of proteome changes, A549-ACE2 cells were pretreated for 6 h with vehicle (PBS) or 0.75 µM DZNep and infected with SARS-CoV-2 and SARS-CoV at MOI 3 for 24 h. Experiment involving tubercidin was performed and analyzed in an analogous manner with the following experimental modifications: 1 µM tubercidin was used with DMSO as vehicle, 3 h pretreatment, SARS-CoV-2 MOI 0.1, SARS-CoV MOI 0.01. Cells were then lysed in SDC lysis buffer (100 mM Tris–HCl pH 8.5; 4% SDC). The following conditions were considered: vehicle-treated uninfected (3 replicates, 4 in tubercidin treatment), DZNep-treated uninfected (4 replicates), vehicle-treated SARS-CoV-2-infected (4 replicates), DZNep-treated SARS-CoV-2-infected (4 replicates), vehicle-treated SARS-CoV-infected (4 replicates), and DZNep-treated SARS-CoV-infected (4 replicates) cells. For the determination of proteome changes in NHBEs, pretreated for 6 h with vehicle (PBS) or 1.5 µM DZNep and infected with SARS-CoV-2 and SARS-CoV at MOI 3 for 24 h, cells were lysed in SDC lysis buffer (100 mM Tris–HCl pH 8.5; 4% SDC). The following conditions were considered: vehicle-treated uninfected, DZNep-treated uninfected, vehicle-treated SARS-CoV-2-infected, DZNep-treated SARS-CoV-2-infected, vehicle-treated SARS-CoV-infected, and DZNep-treated SARS-CoV-infected cells. Cells from four distinct donors were used. Sample preparation was performed as described previously (Stukalov et al, 2021). In brief, protein concentrations of cleared lysates were normalized and 50 µg was used for further processing. To reduce and alkylate proteins, samples were incubated for 5 min at 45°C with TCEP (10 mM) and CAA (40 mM). Samples were digested overnight at 37°C using trypsin (1:100 w/w, enzyme/protein, Sigma-Aldrich) and LysC (1:100 w/w, enzyme/protein, Wako). Resulting peptide solutions were desalted using SDB-RPS StageTips (Empore). Samples were diluted with 1% TFA in isopropanol to a final volume of 200 µl and loaded onto StageTips, and subsequently washed with 200 µl of 1% TFA in isopropanol and 200 µl 0.2% TFA/ 2% ACN. Peptides were eluted with 75 µl of 1.25% ammonium hydroxide ($NH_4OH$) in 80% ACN and dried using a SpeedVac centrifuge (Eppendorf, Concentrator plus). Next, the peptides were reconstituted in buffer A* (0.2% TFA/ 2% ACN) prior to LC–MS/MS analysis. Peptide concentrations were measured optically at 280 nm (Nanodrop 2000, Thermo Scientific) and subsequently equalized using buffer A*. One microgram peptide was subjected to LC–MS/MS, and protein groups were quantified (MaxQuant version 1.6.10.43) with LFQ normalization (A549s) and without LFQ normalization (NHBEs) as described previously (Stukalov et al, 2021).

The analysis of MS datasets was performed using R version 4.0.2. LFQ values were $\log_2$-transformed, and protein groups only identified by site, reverse matches, and potential contaminants were excluded from the analysis. Additionally, protein groups quantified by a single peptide or not detected in all replicates of at least one condition were excluded from further analysis. In NHBE dataset, LFQ values were normalized for donor-specific effects on protein abundance. In short, the protein $\log_2$ intensities were compared across conditions in a donor-wise manner, and systematic deviations across conditions were subtracted in order to get normalized LFQ values.

The imputation of missing $\log_2$ intensity values was done similar to the method implemented in Perseus (Tyanova et al, 2016): The mean and the standard deviation of $\log_2$ intensities were calculated for each dataset, and missing values were replaced by sampling from the normal distribution with the following parameters: 0.3 * standard deviation, mean – 1.8 * standard deviation. In addition, effect scaling was performed using the Gaussian generalized linear modeling approach (core function glm) to allow for quantitative comparison between virus infections and treatments in different

contexts. In short, the following experiment design was used: norm. log$_2$-LFQ ~ virus + virus:treatment, where virus refers to infection with mock, SARS-CoV, or SARS-CoV-2, and treatment refers to vehicle or DZNep treatment. Median absolute values of significant effects ($P < 0.01$) originating from virus and virus:treatment coefficients were calculated and divided by median of SARS-CoV-2 and mock:DZNep, respectively, resulting in coefficient $1 \pm 0.15$ that were used in downstream analysis as coefficients in experimental design matrix.

The following experiment design was used for LASSO-based differential protein abundance analysis: LFQ ~ virus + virus:treatment, where virus refers to infection with mock-, SARS-CoV, or SARS-CoV-2, and treatment refers to vehicle or DZNep treatment. The following effects were thus estimated: effect of SARS-CoV infection, effect of SARS-CoV-2 infection, effect of DZNep treatment of mock-infected cells, effect of DZNep treatment of SARS-CoV-infected cells, and the effect of DZNep treatment of SARS-CoV-2-infected cells. The estimation of LASSO model parameters was performed using R package glmnet (Friedman *et al*, 2010; Simon *et al*, 2011) (version 4.0.2) with thresh = 1e-28, maxit = 1e7, and nfolds = 11. The exact model coefficients and lambda value at cross-validation minimum (lambda.min) were extracted and used for *P*-value estimation by fixed-lambda LASSO inference using the R package selectiveInference (Lee *et al*, 2013), version 1.2.5. Default parameters were used with the following modifications: tol.beta = 0.025, alpha = 0.1, tailarea_rtol = 0.1, tol.kkt = 0.1, and bits = 100. The bits parameter was set to 300 or 500 if the convergence was not reached. The sigma was explicitly estimated using function estimateSigma from the same package. No multiple hypothesis *P*-value correction was performed since that is facilitated by the choice of lambda. The following thresholds were applied to LASSO analysis results to identify statistically significant effects (log$_2$ fold changes): $P < 10^{-5}$ and abs (log$_2$ fold change) $> 0.5$ for the NHBE data, and $P < 10^{-4}$ and abs (log$_2$ fold change) $> 0.2$ for A549 data. If a protein reached significance in one infected condition, or one treated condition, and not others, the significance thresholds for the other conditions were relaxed to: $P < 10^{-2}$ and abs (log$_2$ fold change) $> 0.2$, in order to avoid over-estimating differences among similar infections or drug treatments.

Protein GO-term annotations were retrieved using R package biomaRt (Durinck *et al*, 2009; version 2.45.5). Fisher's exact test was employed, and FDR-adjusted *P*-values were used to identify the terms that are significantly enriched among the changing proteins (threshold: $P < 10^{-2}$).

Proteins, significantly changing in the same direction (up- or downregulated) upon DZNep treatment of SARS-CoV- and SARS-CoV-2-infected NHBEs as determined by the above described analysis (marked in gray and dark-gray in Fig 4B), were used in network diffusion analysis. Network diffusion analysis was performed using ReactomeFI network v2019 (Wu *et al*, 2014). Random walk with restart kernel (R) was computed for this network in undirected manner, with restart probability of 0.4 according to the following equation: R = alpha * (I − (1−alpha)*W)$^{-1}$, where I is the identity matrix, and W is the weight matrix computed as W = D$^{-1}$ * A, where D is degree diagonal matrix, and A is adjacency matrix for ReactomeFI graph. The diagonal values of the R matrix, representing restart and feedback flows, were excluded from subsequent analysis and set to 0. The significant hits from MS data analysis

were mapped to genes in the ReactomeFI network by matching gene names or their synonyms (from the biomaRt_hsapiens gene ensemble dataset) with the gene names in ReactomeFI. Nodes with significant flows originating from nodes representing hits in individual analyses were estimated using a randomization-based approach. All hits and non-hits of the analysis were attributed equal weight (1 and 0, respectively) in subsequent statistical analysis. Flows to all nodes in the network were computed by multiplying the R matrix with the vector of hits described above. Furthermore, nodes in the network were assigned to 8 bins of approximately equal size according to the node degree. The same procedure of calculating inbound flows to all network nodes was repeated for 2,500 iterations, each time using the same number of randomly selected decoy hits from sets of nodes with 1 bin higher node degree (on per-hit basis). The *P*-values describing the significance of functional connectivity to input hits were computed for each node according to the following formula: $P$ = N (iteration with equal or higher inbound flux)/N (iterations). For visualization purposes, the ReactomeFI network was filtered for nodes that were either representing input proteins or proteins with $P < 0.005$ and further trimmed by removing non-hit nodes with degree equal to 1.

### Virus adaptation and competition assays

Vero E6 cells were seeded in T-175 flasks at 15 million cells per flask 24 h before the standard culturing medium was exchanged to one including treatments of choice (0.025% DMSO, 1.25 μM DZNep or 2.5 μM FIDAS-5). Six hours post-treatment, the cultures were inoculated with SARS-CoV-2-MUC-IMB-1 (P0) at MOI 0.01. Forty-eight hours post-inoculation, the supernatant was harvested, spun at 1,000 *g* for 5 min, and further processed for RNA isolation and titration of infectious viral particle content as described above. Deduced titers were used to inoculate freshly prepared Vero E6 cells as described for P0. The process was repeated until reaching passage 10 and is schematically depicted in Fig 5A. Isolated viral genomic RNA was reverse-transcribed as described above and submitted for sequencing (described below).

For pairwise comparison of replication fitness (competition assay), Vero E6 cells were seeded in 12-well plate at a density of 0.2 million cells per well 24 h before the standard culturing medium was exchanged for one containing treatments of choice (0.025% DMSO, 1.25 μM DZNep or 2.5 μM FIDAS-5). Six hours post-treatment, the cultures were inoculated with 1 to 1 mixture (according to infectious particle content) of (i) DMSO P10 and DZNep P10, (ii) DMSO P10 and FIDAS-5 P10, and (iii) DMSO P10 and P0, at MOI 0.01. A part of the inoculum was saved for sequencing analysis. Forty-eight hours post-infection, the culture supernatant was harvested and its RNA content isolated, which was further reverse-transcribed as described above and submitted for sequencing (described below).

For sequencing, SARS-CoV-2 genomes were prepared from amplicon pools, generated with a balanced primer pool according to ARTICv3 protocol (DNA Pipelines R&D *et al*, 2020). Amplicons were converted to barcoded Illumina sequencing libraries with the Nextera XT kit (Illumina, San Diego, USA) in a miniaturized version using a Mantis dispenser (Formulatrix, Bedford, USA) and sequenced on an Illumina NextSeq1000. The obtained sequence reads were demultiplexed and aligned to the SARS-CoV-2 reference

genome (NC 045512.2) with BWA-MEM (preprint: Li, 2013). The read depth along the reference genome was calculated with samtools depth. Variants were called using Freebayes (Cingolani *et al*, 2012) using a ploidy of 1 (−p 1). The effects of genetic variants on amino acid sequences were predicted with SnpEff (Cingolani *et al*, 2012). The pileups files were generated using samtools (Li *et al*, 2009) and used for consensus sequence generation within the iVar (Grubaugh *et al*, 2019) package with default settings. Multiple sequence alignments of the consensus sequences were calculated using MAFFT (v7.475; Katoh & Standley, 2013), which were passed to IQ-TREE2 (v.2.1.2; Minh *et al*, 2020) to calculate the Newick tree.

Ratios between viral isolates in virus competition assay were calculated using mutations depicted in Fig EV5G according to the following formulas, assuming no adaptation events during the course of virus competition assay:

$$S_{vf} = X \times A_{vf} + Y \times B_{vf}$$

$$X + Y = 1,$$

where $S_{vf}$, $A_{vf}$, and $B_{vf}$ are measured variation frequencies of particular mutation in sample of interest, isolate $A$ used in inoculum, and isolate $B$ used in inoculum, respectively. $X$ and $Y$ describe the ratios of isolate $A$ and isolate $B$ in the sample, respectively, and in sum equal to unity. $X$ and $Y$ can be deduced from equations above in the following manner:

$$X = \frac{S_{vf} - B_{vf}}{A_{vf} - B_{vf}}$$

$$Y = 1 - X$$

### Quantitative analysis of co-treatments

Viral inhibition assays utilizing DZNep and remdesivir (or IFN-α) co-treatment and SARS-CoV-2-GFP virus were performed as described above. For remdesivir, A549-RFP-ACE2 cell line was used and a number of RFP-positive cell nuclei were used for normalization of virus reporter signal (instead of phase-based cell confluence used for IFN-α). The calculations of combination indexes for mutually exclusive drugs were performed as described previously (Chou & Talalay, 1984). In short, fractions of system affected and unaffected ($f_a$ and $f_u$, respectively) were calculated for means of normalized GFP integrated intensities (NGII) originating from individual treatment conditions according to the following equation:

$$f_a = 1 - f_u = 1 - \frac{NGII\left(c_{DZNep}, c_{Remdesivir}\right)}{NGII\left(vehicle, vehicle\right)}$$

Half-maximal effective concentrations (EC50) and Hill-type coefficients (m) were calculated by performing linear modeling of vehicle-co-treated data according to the following equations:

$$\log_2\left(\frac{f_a\left(c_{DZNep}, vehicle\right)}{1 - f_a\left(c_{DZNep}, vehicle\right)}\right) = m \times \log_2 c_{DZNep} - m \\ \times \log_2 EC50_{DZNep}$$

$$\log_2\left(\frac{f_a\left(vehicle, c_{Remdesivir}\right)}{1 - f_a\left(vehicle, c_{Remdesivir}\right)}\right) = m \times \log_2 c_{Remdesivir} - m \\ \times \log_2 EC50_{Remdesivir}$$

Combination index (CI) was further calculated according to the following equations:

$$D^t_{DZNep} = EC50_{DZNep} \times \left(\frac{f_a}{1 - f_a}\right)^{1/m_{DZNep}}$$

$$D^t_{Remdesivir} = EC50_{Remdesivir} \times \left(\frac{f_a}{1 - f_a}\right)^{1/m_{Remdesivir}}$$

$$IC = \frac{c_{DZNep}}{c^t_{DZNep}} + \frac{c_{Remdesivir}}{c^t_{Remdesivir}}$$

### *In vivo* experiments

Eight- to 10-week-old female C57BL/6J mice were purchased from Charles River Laboratories. Mice were anesthetized with 90 mg/kg ketamine (WDT) and 9 mg/kg xylazine (Serumwerk Bernburg AG). Mice were inoculated intranasally with $2.5 \times 10^2$ pfu of SARS-CoV-2 beta variant (also known as B.1.351). Infected mice were intranasally treated with 25 μg of tubercidin or 10 μg of DZNep at 30–60 min and 24 h post-infection. All animal experiments using SARS-CoV-2 were performed in a biosafety level 3 facility at University Hospital Bonn according to institutional and governmental guidelines of animal welfare (animal experiment application number: 81-02.04.2019.A247).

### Quantification of virus transcripts in mouse lung material by RT–qPCR

At 2 days post-infection, lungs of infected mice were harvested and homogenized in TRIzol (Invitrogen) using gentleMACS Octo Dissociator (Miltenyi Biotec). RNA was extracted from the homogenates following the manufacturer's protocol. cDNA was generated using High-Capacity cDNA Reverse Transcription Kit (Applied Biosystems). To quantify the viral RNA, real-time quantitative PCR was performed by Step One Plus Real-Time PCR System (Applied Biosystems) using Fast SYBR Green Master Mix (Applied Biosystems) and TaqMan Fast Advanced Master Mix (Applied Biosystems; for transcripts *M*, *E*, and *18s rRNA*), and by QuantStudio 3 Real-Time PCR system (Thermo Fisher) using PowerUp SYBR Green (Thermo Fisher; for transcripts *N* and *Actb*). RT–qPCR primers were designed for SARS-CoV-2 genes as below: 5′-TGTGACATCAAGGACCTGCC-3′ and 5′-CTGAGTCACCTGCTACACGC-3′ for SARS-CoV-2 M; 5′-ACAGGTACGTTAATAGTTAATAGCGT-3′ and 5′-ATATTGCAGCAGTACGCACACA-3′ for SARS-CoV-2 E; and 5′-TTACAAACATTGGCCGCAAA-3′ and 5′-GCGCGACATTCCGAAGAA-3′ for SARS-CoV-2 N. Levels of viral transcripts *M* and *E* were normalized with *18s rRNA* levels using the TaqMan probe for eukaryotic 18s rRNA (Hs99999901_s1, Applied Biosystems). Levels of viral transcript *N* were normalized with *Actb* levels (RT–qPCR primers: 5′-CTCTGGCTCCTAGCACCATGAAGA-3′ and 5′-GTAAAACGCAGCTCAGTAACAGTCCG-3′).

**Quantification of viral load in mouse lung material by plaque assay**

Thirty milligram of lungs was collected from infected mice at 2 days post-infection. Lungs were homogenized in 300 μl of PBS using Tissue Grinder Mixy Professional (NIPPON Genetics EUROPE, NG010). Homogenates were cleared by centrifugation twice (200 *g*, 5 min, 4°C; and 20,000 *g*, 5 min, 4°C), and the supernatants were stored at −80°C until further processing. The viral titers were determined by the plaque assay using Vero E6 cells as described above.

# Data availability

The mass spectrometry proteomics data have been deposited to the ProteomeXchange Consortium via the PRIDE (Perez-Riverol *et al*, 2022) partner repository with the dataset identifier PXD034361 (http://www.ebi.ac.uk/pride/archive/projects/PXD034361).

**Expanded View** for this article is available online.

## Acknowledgements

We thank Janett Wieseler, Dr. Yueyuan Hu, and Dr. Alexander Herrmann for excellent technical assistance and Volker Thiel for the SARS-CoV-2 GFP virus. We further acknowledge the Microscopy Core Facility (MCF) of the Medical Faculty at the University of Bonn and BayBioMS@MRI for support. This project was supported by the European Research Council (ERC-CoG, Grant nr. 817798), Bavarian State Ministry of Science and Arts (Bavarian Research Network FOR-COVID) and the Helmholtz Association's Initiative and Networking Fund (KA1-Co-02 "COVIPA") to APic, the Federal Ministry for Education and Research (BMBF; COVINET to AP, TTU 01.810 to BMK), the German Research Foundation (DFG; Germany's Excellence Strategy—EXC 2151—390873048 to HK; TRR237 Grant No. 369799452 to APic (A07), BMK (A04), and HK (B22); and TRR179 (TP11), PI 1084/4, PI 1084/5, and PI 1084/7 to APic), CAJ, MO, and CBS-W were supported by funding from the German Center of Lung Research (DZL). Open Access funding enabled and organized by ProjektDEAL.

## Author contributions

**Valter Bergant:** Conceptualization; supervision; investigation. **Shintaro Yamada:** Investigation. **Vincent Grass:** Investigation. **Yuta Tsukamoto:** Investigation. **Teresa Lavacca:** Investigation. **Karsten Krey:** Investigation. **Maria-Teresa Mühlhofer:** Investigation. **Sabine Wittmann:** Investigation. **Armin Ensser:** Investigation. **Alexandra Herrmann:** Investigation. **Anja Vom Hemdt:** Investigation. **Yuriko Tomita:** Investigation. **Shutoku Matsuyama:** Resources. **Takatsugu Hirokawa:** Formal analysis; visualization. **Yiqi Huang:** Investigation. **Antonio Piras:** Investigation. **Constanze A Jakwerth:** Resources. **Madlen Oelsner:** Resources. **Susanne Thieme:** Formal analysis; visualization. **Alexander Graf:** Formal analysis; visualization. **Stefan Krebs:** Resources. **Helmut Blum:** Resources. **Beate M Kümmerer:** Resources. **Alexey Stukalov:** Formal analysis; visualization. **Carsten B Schmidt-Weber:** Resources. **Manabu Igarashi:** Formal analysis; visualization. **Thomas Gramberg:** Resources. **Andreas Pichlmair:** Conceptualization; supervision; funding acquisition; investigation. **Hiroki Kato:** Conceptualization; supervision; funding acquisition; investigation.

In addition to the CRediT author contributions listed above, the contributions in detail are:

HK, APic, and VB conceptualized, supervised, and investigated the study. HK and APic acquired funding. TH, MI, ST, AG, and AS performed formal analysis

and visualized the study. SY, YTs, SW, AE, AH, AVH, YTo, VG, TL, KK, M-TM, YH, and APir investigated the study. SM, BMK, TG, CAJ, MO, SK, HB, and CBS-W provided resources.

## Disclosure and competing interests statement

VB, VG, and APic are co-inventors on the patent application related to the content of this manuscript.

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
