## [Review Process File · The EMBO Journal]

Attenuation of SARS-CoV-2 replication and associated inflammation by concomitant targeting of viral and host cap 2'-O-ribose methyltransferases

Valter Bergant, Shintaro Yamada, Vincent Grass, Yuta Tsukamoto, Teresa Lavacca, Karsten Krey, Maria-Teresa Mühlhofer, Sabine Wittmann, Armin Ensser, Alexandra Herrmann, Anja Hemdt, Yuriko Tomita, Shutoku Matsuyama, Takatsugu Hirokawa, Yiqi Huang, Antonio Piras, Constanze Jakwerth, Madlen Oelsner, Susanne Thieme, Alexander Graf, Stefan Krebs, Helmut Blum, Beate Kümmerer, Alexey Stukalov, Carsten Schmidt-Weber, Manabu IGARASHI, Thomas Gramberg, Andreas Pichlmair, and Hiroki Kato

DOI: 10.15252/emj.2022111608

Corresponding author(s): Andreas Pichlmair (andreas.pichlmair@tum.de) , Hiroki Kato (hkato@uni-bonn.de), Andreas Pichlmair (andreas.pichlmair@tum.de)

Review Timeline:

Submission Date:	8th May 22
Editorial Decision:	25th May 22
Revision Received:	13th Jun 22
Editorial Decision:	23rd Jun 22
Revision Received:	23rd Jun 22
Accepted:	27th Jun 22

Editor: Karin Dumstrei

Transaction Report:

(Please note that the manuscript was previously reviewed at another journal and the reports were taken into account in the decision making process at The EMBO Journal. Since the original reviews are not subject to EMBO's transparent review process policy, the reports and author response cannot be published)

Note: With the exception of the correction of typographical or spelling errors that could be a source of ambiguity, letters and reports are not edited. Depending on transfer agreements, referee reports obtained elsewhere may or may not be included in this compilation. Referee reports are anonymous unless the Referee chooses to sign their reports.)

Dear Andreas,

Thank you for submitting your manuscript to The EMBO Journal. This manuscript has been reviewed at another journal and transferred with referee reports and point-by-point response to The EMBO Journal.

I involved an arbitrating referee who looked at the manuscript and point-by-point response. As you can see below the referee finds the analysis insightful and important. The referee raises a few issues that would be good to sort out.

When you submit the revised version will you also please take care of the following issues:

- Upload a word doc of the text file
- We are missing 3-5 keywords
- We need a Author Contribution section
- Competing Interests should be called Disclosure and Competing Interests Statement.
- We need an author checklist - see also guide to authors
- We generally only allow 3 1st authors - it is not clear from the legend if you have 2 or 4.
- The reference list needs to be updated to EMBO Journal style.
- Regarding the figures: we no longer have supplemental figures, but instead have expanded view figures that are typeset and integrated into the html file. We allow only 5 expanded view figures and the rest would have to be added to an appendix. The appendix would also need a ToC. Alternatively, you can turn the supplementary figures into regular figures. See also our guide to authors or contact me if you need more help.
- Some tables are too big and will not convert to pdf. They should be changed to Datasets (this way they will stay as excel files) and called out as Dataset EV1 with legends in a separate tab within the xlsx sheet. See also our guide to authors or contact me if you need more help.
- Please double check that the funding is also entered into the online submission system =>Association's Initiative and Networking Fund (KA1-Co-02 "COVIPA"),BMK (A04) and HK (B22), TRR179 (TP11), PI 1084/4, PI 1084/5 and PI 1084/7 are missing from EJP system.
- We need a Data Availability section. This is the place to enter accession numbers etc. It should be place after the Materials and methods and before Acknowledgements. If no data needs to deposited in a database please add: This study includes no data deposited in external repositories.
- We include a synopsis of the paper that is visible on the html file (see <http://emboj.embopress.org/>). Can you provide me with a general summary statement and 3-5 bullet points that capture the key findings of the paper?
- It would also be good if you could send me a summary figure for the synopsis. The size should be 550 wide by 400 high (pixels).

That should be all! Let me know if you have any further questions

Best

Karin

Karin Dumstrei, PhD
Senior Editor
The EMBO Journal

Guide For Authors: <https://www.embopress.org/page/journal/14602075/authorguide>

Use the link below to submit your revision:

Referee #1:

I have now read the manuscript as well as the rebuttal letter co-submitted by the authors.

Overall I find this as a strong and important paper but of course it has lost a bit of its flashing novelty due to other recent published papers.

Nevertheless, I do endorse it for publication and I do not agree to the critics raised by reviewer 2.

The authors have addressed the comments by Reviewer 1 and 3 appropriately and I really enjoyed the flow the paper was written in and the overall conclusions.

This here is a school book example of how we can exploit the concept of repurposing drugs developed for other modalities and by using novel virological mechanistic understanding we can find drugs applicable to treat a new emerging disease.

I have some minor issues with data missing in the manuscript that should be addressed:

line 175 -> the MOI used is extremely low. Would be good to include a statement for why this approach was used -> or confirm data with a higher MOI

line 233 -> will be important to show ICE analysis and WB results for the proposed MTr1 KO cell line

line 272-274 -> will be important to show ICE analysis and WB results for the proposed KO cell lines

Figure 2 -> lack stats on panel c, d and e

Referee #1

I have now read the manuscript as well as the rebuttal letter co-submitted by the authors.

Overall I find this as a strong and important paper but of course it has lost a bit of its flashing novelty due to other recent published papers.

Nevertheless, I do endorse it for publication and I do not agree to the critics raised by reviewer 2.

The authors have addressed the comments by Reviewer 1 and 3 appropriately and I really enjoyed the flow the paper was written in and the overall conclusions.

This here is a school book example of how we can exploit the concept of repurposing drugs developed for other modalities and by using novel virological mechanistic understanding we can find drugs applicable to treat a new emerging disease.

We thank the reviewer for reviewing our work and recognizing the value in it.

I have some minor issues with data missing in the manuscript that should be addressed:

line 175 -> the MOI used is extremely low. Would be good to include a statement for why this approach was used -> or confirm data with a higher MOI

We thank the reviewer for this comment and recognize its importance. Many compounds exhibit best antiviral efficacy upon infection with low virus dose and show diminished efficacy upon increasing MOIs. We therefore chose to use low MOI infections for antiviral efficacy screenings. We demonstrate the antiviral efficacy of Tubercidin upon infections with wild-type SARS-CoV-2 employing MOI 0.1 (versus 0.01 as used in the experiment related to this comment) in Figure 1f, g. Figure 3f furthermore depicts results obtained from treating cells upon infection with SARS-CoV-2-GFP reporter virus at MOI 3.

line 233 -> will be important to show ICE analysis and WB results for the proposed MTr1 KO cell line

We thank the reviewer for this comment and provide WB analysis in Figure EV1j.

line 272-274 -> will be important to show ICE analysis and WB results for the proposed KO cell lines

We thank the reviewer for this comment and provide RT-qPCR analysis of gene expression in Figure EV2a.

Figure 2 -> lack stats on panel c, d and e

In Figure 2e, the high amount of missing values due to the detection limit of the plaque assay impedes statistical handling of the data. The biological difference is nevertheless clear, and the lack of statistics does not impair the interpretation of the panels. For consistency reasons, we left out the statistics also in Figure 2d.

In Figure 2c, we could not observe any biologically meaningful trends in the data, even though some differences may be statistically significant (due to low data scattering, etc.). Adding statistics to this panel may be misleading to the readers, if misinterpreted as something biologically relevant. For this reason, we ask the reviewer for understanding to leave out the statistical analysis for this particular panels.

Hi Andreas,

Thanks for sending me the revised version. I have looked at it and all looks good. I am therefore pleased to let you know that I will accept the manuscript for publication here. Before sending you the formal acceptance letter there are just a few formatting issues that we need to sort out:

Reference list: Please reduce the number of authors in the Perez-Riverol et reference to 10 authors et al.

We don't allow Data Not Shown (page 8) - please rephrase

We are missing the ORCID ID for Kato

Please include the funding information in the Acknowledgements.

The legends for the dataset EV tables need to be removed from the manuscript file and added as a separate tab in the excel files.

The Appendix needs a ToC and the figure legend removed from the MS and added to the Appendix file.

Please double check that the MS sections are in the right order

Correct "Summary" needs correcting to 'Abstract'.

Correct "Figure text" to 'Figure Legends' and "Expanded View Figure Text" to 'Expanded View Figure Legends'.

Please make sure to make the dataset PXD034361 is made public

While checking the SD I have noted that for Figure EV1 that the the second row of images in the figure panel FEV1h is missing (the 3 images A549 + IFN-alpha). I am just pointing this out in case it was overlooked.

Our publisher has also done their pre-publication check on your manuscript. When you log into the manuscript submission system you will see the file "Data Edited Manuscript file". Please take a look at the word file and the comments regarding the figure legends and respond to the issues.

Please upload a clean MS version and a point-by-point response to the editorial points. You can remove synopsis text from the MS as it is uploaded as a separate file.

Once I get the revised version in then I will move forward with the acceptance of the MS for publication here.

With best wishes

Karin

Karin Dumstrei, PhD
Senior Editor
The EMBO Journal

link to submit your revision:

Thanks for sending me the revised version. I have looked at it and all looks good. I am therefore pleased to let you know that I will accept the manuscript for publication here. Before sending you the formal acceptance letter there are just a few formatting issues that we need to sort out:

Reference list: Please reduce the number of authors in the Perez-Riverol et reference to 10 authors et al.

Done.

We don't allow Data Not Shown (page 8) - please rephrase

The phrase was removed.

We are missing the ORCID ID for Kato

Done.

Please include the funding information in the Acknowledgements.

Done.

The legends for the dataset EV tables need to be removed from the manuscript file and added as a separate tab in the excel files.

Done.

The Appendix needs a ToC and the figure legend removed from the MS and added to the Appendix file.

Appendix table of content added as a separate file.

Please double check that the MS sections are in the right order

Done. Some sections were moved to follow section order as specified in author guidelines.

Correct "Summary" needs correcting to 'Abstract'.

Done.

Correct "Figure text" to 'Figure Legends' and "Expanded View Figure Text" to 'Expanded View Figure Legends'.

Done.

Please make sure to make the dataset PXD034361 is made public

We submitted the request to PRIDE – the dataset should become accessible in the next few days.

While checking the SD I have noted that for Figure EV1 that the the second raw of images in the figure panel FEV1h is missing (the 3 images A549 + IFN-alpha). I am just pointing this out in case it was overlooked.

We apologise, this was overlooked. Was added.

Our publisher has also done their pre-publication check on your manuscript. When you log into the manuscript submission system you will see the file "Data Edited Manuscript file". Please take a look at the word file and the comments regarding the figure legends and respond to the issues.

Done.

Please upload a clean MS version and a point-by-point response to the editorial points. You can remove synopsis text from the MS as it is uploaded as a separate file.

Done.

Dear Andreas and Hiroki,

Thank you for submitting the revised version to The EMBO Journal. I have now had a chance to take a look at it and all looks good.

I am therefore very pleased to accept the MS for publication in The EMBO Journal.

Congratulations on a nice study!

With best wishes

Karin

Karin Dumstrei, PhD
Senior Editor
The EMBO Journal

Please note that it is EMBO Journal policy for the transcript of the editorial process (containing referee reports and your response letter) to be published as an online supplement to each paper. If you do NOT want this, you will need to inform the Editorial Office via email immediately. More information is available here:

<https://www.embopress.org/page/journal/14602075/authorguide#transparentprocess>

Your manuscript will be processed for publication in the journal by EMBO Press. Manuscripts in the PDF and electronic editions of The EMBO Journal will be copy edited, and you will be provided with page proofs prior to publication. Please note that supplementary information is not included in the proofs.

You will be contacted by Wiley Author Services to complete licensing and payment information. The required 'Page Charges Authorization Form' is available here: https://www.embopress.org/pb-assets/embo-site/tej_apc.pdf - please download and complete the form and return to embopressproduction@wiley.com

Should you be planning a Press Release on your article, please get in contact with embojournal@wiley.com as early as possible, in order to coordinate publication and release dates.

If you have any questions, please do not hesitate to call or email the Editorial Office. Thank you for your contribution to The EMBO Journal.
